

# A long-term dataset of climatic mass balance, snow conditions and runoff in Svalbard (1957-2018)

Ward van Pelt[1], Veijo Pohjola[1], Rickard Pettersson[1], Sergey Marchenko[1], Jack Kohler[2], Bartek Luks[3], Jon Ove Hagen[4], Thomas V. Schuler[4,5], Thorben Dunse[4,6], Brice Noël[7], and Carleen Reijmer[7]

[1]Department of Earth Sciences, Uppsala University, Uppsala, Sweden
[2]Norwegian Polar Institute, Tromsø, Norway
[3]Institute of Geophysics, Polish Academy of Sciences, Warszawa, Poland
[4]Department of Geosciences, University of Oslo, Oslo, Norway
[5]Department of Arctic Geophysics, University Centre in Svalbard, Longyearbyen, Svalbard
[6]Department of Environmental Sciences, Western Norway University of Applied Sciences, Norway
[7]Institute for Marine and Atmospheric Research Utrecht, Utrecht University, Utrecht, The Netherlands

**Correspondence:** Ward van Pelt (ward.van.pelt@geo.uu.se)

**Abstract.** The climate in Svalbard is undergoing amplified change compared to the global mean. This has major implications for runoff from glaciers and seasonal snow on land. We use a coupled energy balance − subsurface model, forced with downscaled regional climate model fields, and apply it to both glacier-covered and land areas in Svalbard. This generates a long-term (1957-2018) distributed dataset of climatic mass balance (CMB), snow conditions and runoff with a $1 \times 1$-km spatial and 3-hourly temporal resolution. Observational data including stake measurements, automatic weather station data and subsurface data across Svalbard are used for model calibration and validation. We find a weakly positive mean CMB ($+0.09$ m w.e. a$^{-1}$) over the simulation period, which only fractionally compensates for mass loss through calving. Pronounced warming and a weak precipitation increase lead to a spatial-mean negative CMB trend ($-0.06$ m w.e. a$^{-1}$ decade$^{-1}$), and an increase in the equilibrium line altitude (ELA) by 17 m decade$^{-1}$, with largest changes in southern and central Svalbard. The retreating ELA in turn causes firn air volume to decrease by 4% decade$^{-1}$, which, in combination with winter warming induces a substantial reduction of refreezing in both glacier-covered and land areas (average $-4\%$ decade$^{-1}$). A combination of increased melt and reduced refreezing cause glacier runoff (average 34.3 Gt a$^{-1}$) to double over the simulation period, while discharge from land (average 10.6 Gt a$^{-1}$) remains nearly unchanged. As a result, the relative contribution of land runoff to total runoff drops from 30 to 20% during 1957-2018. Seasonal snow on land and in glacier ablation zones is found to arrive later in autumn ($+1.4$ days decade$^{-1}$), while no significant changes occurred in the date of snow disappearance in spring/summer. Altogether, the output of the simulation provides an extensive dataset that may be of use in a wide range of applications ranging from runoff modelling to ecosystem studies.

## 1 Introduction

The Arctic climate is changing at a faster rate than the global mean (IPCC, 2014; AMAP, 2017) as a result of climate feedbacks triggered by changing sea-ice cover (Serreze and Barry, 2011; Bintanja and Van Der Linden, 2013). The climate in Svalbard,





located at the southwestern boundary of wintertime sea-ice and at the northeastern end of the North Atlantic Drift, is primarily controlled by sea-ice cover trends (Divine and Dick, 2006; Day et al., 2012) and trends in prevailing wind direction (Hanssen-Bauer and Førland, 1998; Lang et al., 2015). The homogenized observational air temperature time-series from Longyearbyen (1898−2012) reveal a linear trend of 2.6 $^o$C per century, with three−four times stronger warming in winter/spring than in sum-

mer (Nordli et al., 2014). Longterm precipitation records in Svalbard are uncertain due to the local character of measurements and instrumental errors (Førland and Hanssen-Bauer, 2000; Førland et al., 2011), but show an overall increase that is coherent with large-scale Arctic-wide assessments (e.g., Zhang et al., 2013). Ongoing climate trends strongly affect the state of both glaciers and seasonal snow in Svalbard (e.g., Van Pelt et al., 2016a; Østby et al., 2017).

In response to warming, glaciers in Svalbard with a current estimated volume of ∼6,200 km$^3$ (1.5 cm sea level equivalent;

Fürst et al., 2018), and area of 33,775 km$^2$ ($\sim 57\%$ of the total area of Svalbard; Fig. 1), have in recent decades shrunk by $\sim 80$ km$^2$ a$^{-1}$ (Nuth et al., 2013), primarily due to low-elevation thinning and associated retreat (e.g., Moholdt et al., 2010; Nuth et al., 2012). Total glacier mass balance is the sum of frontal ablation, basal ablation, and the climatic mass balance (CMB), representing the mass change due to atmosphere - surface - snow pack interactions (Cogley et al., 2011). CMB measurements in Svalbard started on Austre Brøggerbreen (since 1967), followed by Midtre Lovénbreen (since 1968), both in northwestern

Svalbard. Since the 1980s, CMB monitoring has extended also to southern, central and northeastern Svalbard (Fig. 1, Table 1). Although a negative trend in CMB is apparent for most observed glaciers, the scarcity of the data in space and time does not allow for a detailed estimation of long-term CMB trends for different regions in Svalbard. To overcome this, CMB models, commonly forced with regional climate model or reanalysis fields, have previously been applied to individual glacier basins (e.g., Möller et al., 2013; Van Pelt et al., 2012; Van Pelt and Kohler, 2015) as well as for all glaciers in Svalbard (e.g., Day

et al., 2012; Lang et al., 2015; Aas et al., 2016; Østby et al., 2017; Möller and Kohler, 2018). The use of different CMB models, climate forcings, model calibration and spatial resolution has resulted in a relatively large spread of multi-decadal Svalbard-wide mean CMB and trends in CMB in available literature. For example, Lang et al. (2015) report a negligible CMB trend for 1979−2013, while Østby et al. (2017) report a strong CMB decline over the same period and the longer period 1957−2014. As a result, despite confirmed significant warming in Svalbard since the 1960's (Nordli et al., 2014), its impact on glacier CMB

remains poorly constrained.

Recent climate warming not only has a major impact on glaciers, but also exerts a strong influence on the state of seasonal snow in the glacier-free parts of Svalbard. Previous work has shown that despite a modest increase in Arctic precipitation in recent decades (Zhang et al., 2013; Bintanja and Selten, 2014), the duration of the snow-free season is increasing and that the area with a permanent snow cover is declining (Van Pelt et al., 2016a). It has also been shown that thick ice layers may

form in snowpacks during winters with heavy rainfall events, thereby limiting reindeer access to food supplies and leading to population declines (Kohler and Aanes, 2009; Hansen et al., 2014). Formation of ice at the base of seasonal snowpacks has been projected to increase in a future climate (Hansen et al., 2011), as the fraction of precipitation falling as rain is rising (Bintanja and Andry, 2017). In situ snow observations by means of probing, snow pits, ground-penetrating radar and remote sensing, have been extensively used to assess local-scale patterns and evolution of seasonal snow in Svalbard (e.g., Hagen et al.,





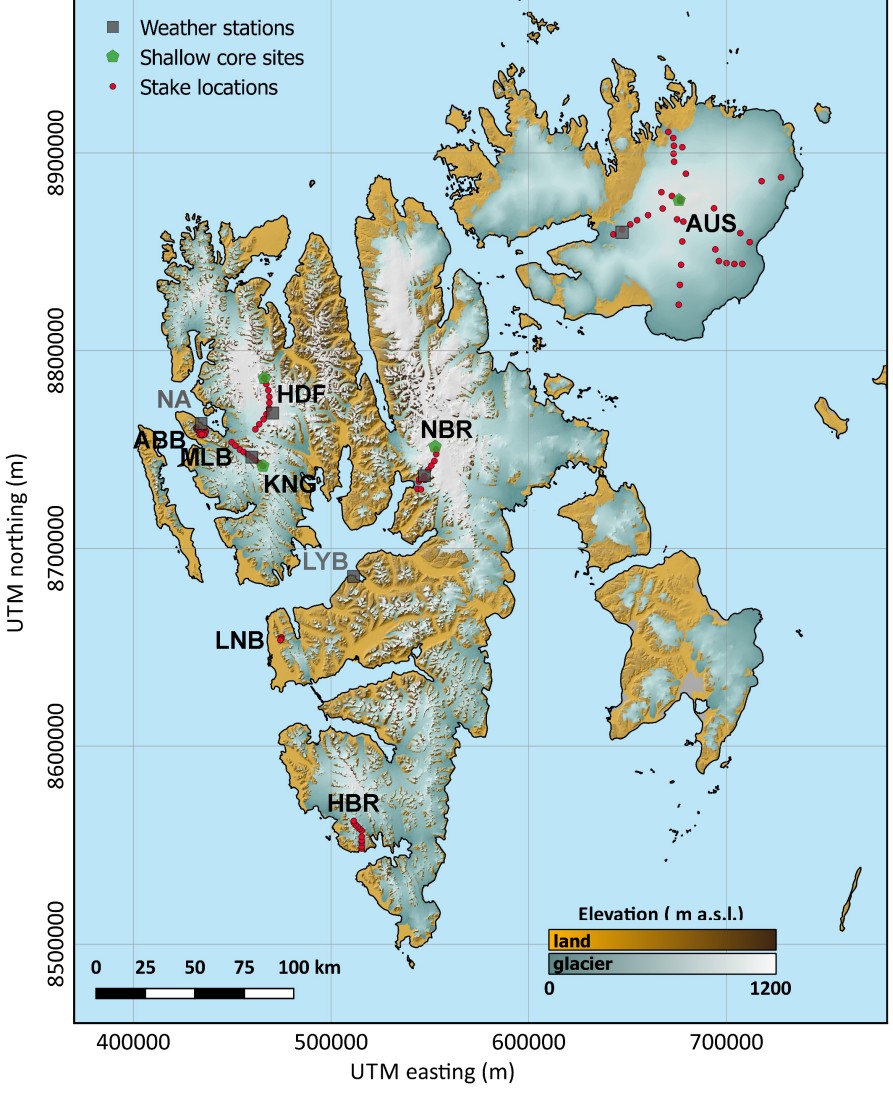

**Figure 1.** Topographic map of Svalbard with different elevation colormaps to distinguish between glacier-covered and land areas. Sites of in situ data collection, including stakes, weather stations and shallow ice cores, are indicated [ABB = Austre Brøggerbreen; AUS = Austfonna; HBR = Hansbreen; HDF = Holtedahlfonna; KNG = Kongsvegen; MLB = Midtre Lovénbreen; LNB = Linnébreen; NBR = Nordenskiöldbreen; LYB = Longyearbyen; NA = Ny-Ålesund]. UTM coordinates in this and later figures are in zone 33 X. The digital elevation model and mask used to produce the map are described in Sect. 2.1, and an overview of the observational data is given in Table 1.

2003; Winther et al., 2003; Van Pelt et al., 2014), but provide only limited insight in snowpack dynamics at large spatial and temporal scales.





In this study, we use a coupled surface energy balance - multilayer subsurface model (Van Pelt et al., 2012, 2016b) and apply it to all of Svalbard to generate a model dataset with a 3-hourly temporal and 1×1-km spatial resolution for the period 1957−2018. In contrast to previous large-scale coupled modelling of glaciers in Svalbard (Lang et al., 2015; Aas et al., 2016; Østby et al., 2017), we apply our model to both glacierized and glacier-free terrain. Furthermore, we implement improved

model physics, and adopt new techniques for climate downscaling and calibration (Sect. 3). Two different model setups are chosen to enable simulating deep subsurface conditions for the glacier-covered part and detailed seasonal snow pack evolution on permafrost for the land part. In situ data of stake mass balance, automatic weather stations and snow conditions (Sect. 2) are used for model calibration and validation (Sect. 3). In Sect. 4 we present and discuss spatial patterns and trends of CMB, snow and firn conditions on glaciers, as well as seasonal snow conditions on land, which allows for a detailed and

unprecedented quantification of seasonal snow and glacier contributions to total discharge from the Svalbard archipelago. The output dataset provides crucial input data for further cryospheric analyses, and may serve as input for studies of marine and terrestrial ecosystems.

## 2 Data

In this section we describe the data used as model input (Sect. 2.1), for model calibration (Sect. 2.2) and for validation of model

results (Sect. 2.3). An overview of all observational data used is given in Table 1.

### 2.1 Input data

A digital elevation model (DEM) with a 20-m spatial resolution, provided by the Norwegian Polar Institute (S0 Terrengmodel Svalbard), has been averaged onto a 1-km resolution grid for the model experiments. Resulting elevations range from sea level to 1552 m a.s.l. (the actual highest point on Svalbard is 1717 m). Glacier outlines were extracted from the GLIMS database

(Global Land Ice Measurements from Space; König et al., 2014) and used to split the terrain into land and glacier-covered areas (Fig. 1), and to estimate equilibrium line altitude for individual glacier basins. Glacier outlines correspond to the period 2001−2010, while the data behind the DEM where collected during 1990−2010. We assume fixed elevations and glacier mask over the simulation period to produce reference surface mass balance (Elsberg et al., 2001).

To generate meteorological forcing fields of air temperature, precipitation, cloud cover, relative humidity and air pressure,

we use 3-hourly output from the High Resolution Limited Area Model (HIRLAM) regional climate model (NORA10 dataset; Norwegian Meteorological Institute; Reistad et al., 2011), covering the period 1957−2018. HIRLAM is forced by European Centre for Medium Range Weather Forecasts (ECMWF) reanalyses (Uppala et al., 2005; Dee et al., 2011). HIRLAM fields with an original 10-km resolution were downscaled to the 1-km model grid resolution using parameter-specific downscaling techniques (Van Pelt et al., 2016a). All meteorological variables were first linearly interpolated onto the 1-km grid, before

additionally applying elevation corrections for temperature (time-dependent lapse rate), precipitation (fixed linear fractional increase with elevation), and air pressure (time-dependent exponential decay with elevation). Average temperature and precipitation, as well as corresponding long-term linear trends are shown in Fig. 2. Throughout the manuscript temporal trends



**Table 1.** Overview of in situ observational data used in this study. The number of stake locations per glacier are indicated in brackets in the second column. [s = summer, w = winter, C = calibration, V = validation, NPI = Norwegian Polar Institute]

| Description | Location | Variables | Period | Frequency | Purpose | Source |
|---|---|---|---|---|---|---|
| Stake measurements | BRG (7x) | $b_s, b_w$ | 1967−2015 | s,w | C | NPI |
| | MLB (4x) | $b_s, b_w$ | 1968−2015 | s,w | C | NPI |
| | KNG (9x) | $b_s, b_w$ | 1987−2015 | s,w | C, V | NPI |
| | HBR (11x) | $b_s, b_w$ | 1989−2012 | s,w | C, V | Polish Acad. of Sciences |
| | HDF (10x) | $b_s, b_w$ | 2003−2015 | s,w | C | NPI |
| | LNB (3x) | $b_s, b_w$ | 2004−2010 | s,w | C | NPI |
| | AUS (27x) | $b_s, b_w$ | 2004−2013 | s,w | C | Univ. of Oslo, NPI |
| | NBR (11x) | $b_s, b_w$ | 2006−2015 | s,w | C | Uppsala & Utrecht Univ. |
| Weather stations | LYB | $T_{air}$ | 1975−2016 | daily | V | Norwegian Meteorol. Inst. |
| | NA | $T_{air}$ | 1969−2015 | daily | V | Norwegian Meteorol. Inst. |
| | AUS | $T_{air}$ | 2004−2016 | daily | V | Univ. of Oslo |
| | KNG | $SW_{net}, T_{air}$ | 2007−2012 | daily | C, V | NPI |
| | HDF | $SW_{net}, T_{air}$ | 2009−2012 | daily | C, V | NPI |
| | NBR | $SW_{net}, T_{air}$ | 2009−2015 | daily | C, V | Uppsala & Utrecht Univ. |
| Shallow cores | KNG | $\rho_{sub}$ | 1996, 2001, 2002, 2007 | - | V | NPI |
| | AUS | $\rho_{sub}$ | 1999, 2008, 2011, 2012 | - | V | Univ. of Oslo, NPI |
| | HDF | $\rho_{sub}$ | 2005, 2008, 2014, 2015 | - | V | NPI |
| | NBR | $\rho_{sub}$ | 2012, 2013, 2014, 2015 | - | V | Uppsala & Utrecht Univ. |

were calculated by means of linear regression of annual time-series; non-significant trends at a 95% confidence interval were set to zero and appear as grey in the associated figures. The long-term mean temperature distribution (Fig. 2a) reveals highest temperatures at low elevation sites in the southwest, and lowest temperatures at high elevations on the Lomonosovfonna ice cap in central Svalbard. Temperature trends are significantly positive for the whole of Svalbard, with the most pronounced trends 5 in the northeast (Fig. 2b). The long-term mean precipitation distribution shows a clear elevation dependence (Fig. 2c), while long-term trends are generally found to be non-significant, except in the north, where there is a significant positive trend (Fig. 2d).

## 2.2 Calibration data

For model calibration, we use records of summer and winter balance ($b_s$, $b_w$) from stake measurements and net shortwave 10 radiation ($SW_{net}$) observed at three automatic weather stations.



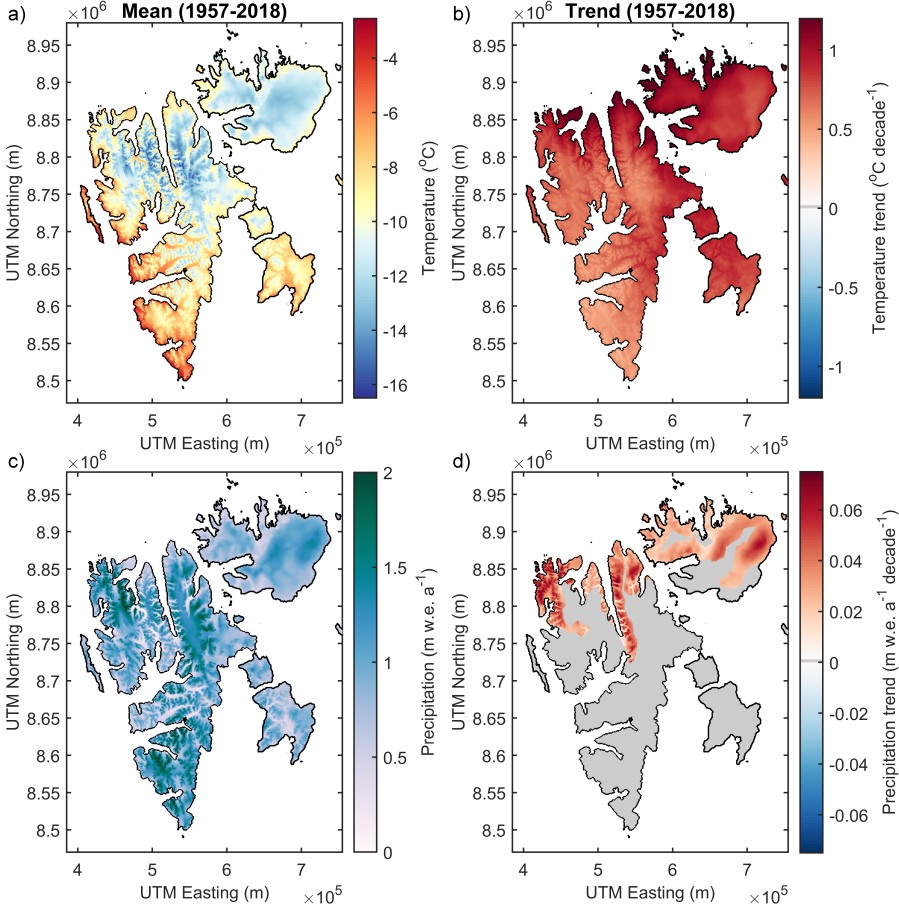

**Figure 2.** Long-term mean air temperature distribution (a) and trends (b). Long-term mean precipitation distribution (c) and trends (d). Non-significant trends at a 95% confidence interval are set to zero (grey).

Stake heights for a set of glaciers around Svalbard (Table 1) are recorded once or twice per year and, in combination with snow density and snow depth data, are converted into summer balance and winter balance estimates. Here, we use data from 82 stake locations in Svalbard, covering eight different glaciers and ice caps (Fig. 1). The Norwegian Polar Institute has collected stake data on a set of glaciers in western Svalbard, including Austre Brøggerbreen (ABB), Midtre Lovénbreen

5  (MLB), Kongsvegen (KNG), Holtedahlfonna (HDF) and Linnébreen (LNB); the oldest record (ABB) dates back to 1967 (e.g., Hagen et al., 1999; Kohler et al., 2007). Stake data on Hansbreen (HBR) have been collected by the Institute of Geophysics, Polish Academy of Sciences since 1989 (Grabiec et al., 2012). The University of Oslo and Norwegian Polar Institute have done stake measurements on Austfonna since 2004 (e.g., Moholdt et al., 2010; Aas et al., 2016). Stake measurements on Nordenskiöldbreen were initiated in 2006 by Uppsala and Utrecht University (e.g., Van Pelt et al., 2012, 2018). Derived net

10  glacier-wide mass balances of ABB, KNG, HDF and HBR are included in the World Glacier Monitoring Service database (WGMS; https://wgms.ch/).





For ABB, MLB and LNB, the dense observation network caused several stake sites to fall within one 1×1-km model grid cell, in which case we only selected the stake location closest to model grid nodes for further comparison with the model results. As a result, we include only four (out of ten) stakes on MLB, seven (out of eleven) on BRG, and three (out of eight) stakes on LNB. The winter balance data for the same set of glaciers were previously described and used in Van Pelt et al. (2016a).

Summer balance is estimated using information of spring (April) and end-of-summer surface height, while spring snow depth is used to distinguish between snow and ice melt. In absence of direct end-of-summer surface height measurements, the depth of the summer surface was inferred from subsequent spring stake height and snow depth data. In the accumulation zone refreezing above the summer surface is accounted for by setting an assumed end-of-summer remaining snow density of 550 kg m$^{-3}$. For calculating summer and winter balance from the model output, we use fixed dates of April 15 and September 1, corresponding to average dates for spring stake data collection and end-of-summer minimum surface height, respectively.

In situ data of $SW_{\mathrm{net}}$ (Table 1), i.e. incoming minus reflected solar radiation, are extracted from radiation measurements at automatic weather stations in central Svalbard (NBR; Van Pelt et al., 2012), and western Svalbard (KNG and HDF; Karner et al., 2013; Van Pelt and Kohler, 2015; Pramanik et al., 2018).

## 2.3 Validation data

In addition to the above-described in situ data used for model calibration, we further use observed density profiles from shallow cores and air temperature time-series observed at (automatic) weather stations for validation of model results.

Shallow cores were drilled during multiple years at four locations in the accumulation zones on KNG (722 m a.s.l.), HDF (1122 m a.s.l.), NBR (1187 m a.s.l.) and AUS (758 m a.s.l.) to obtain density profiles with maximum depths ranging from 7 to 15 m below the surface (Fig. 1; Table 1). For each of the four sites we selected four firn density profiles, collected during different years on NBR (2012, 2013, 2014 and 2015), KNG (1996, 2001, 2002 and 2007), HDF (2005, 2008, 2014 and 2015) and AUS (1999, 2008, 2011 and 2012). Bulk densities are calculated over the full depth of observations and compared to simulated values over the same depth intervals.

We use a combination of air temperature records from the automatic weather stations on the glaciers AUS (Schuler et al., 2014), NBR (Van Pelt et al., 2012), and KNG and HDF (e.g., Karner et al., 2013), as well as from two land-based meteorological stations in Longyearbyen and Ny-Ålesund (data provided by the Norwegian Meteorological Institute through the eKlima data portal) for comparison with downscaled temperatures (Fig. 1; Table 1).

## 3 Model & Setup

### 3.1 Coupled modelling

A coupled modelling system is used to simulate surface and near-surface mass and energy exchange (Van Pelt et al., 2012), which has been used previously to simulate glacier mass balance, (seasonal) snow development and/or runoff in western Svalbard (e.g., Van Pelt and Kohler, 2015; Vallot et al., 2017; How et al., 2017; Winsvold et al., 2018; Pramanik et al., 2018;



Deschamps-Berger et al., 2019), central Svalbard (e.g., Van Pelt et al., 2012, 2014; Vega et al., 2016; Marchenko et al., 2017b; Van Pelt et al., 2018) and on an idealized Svalbard glacier (Van Pelt et al., 2016b). In this study, the model is applied for the first time to the whole of Svalbard. At the surface, an energy balance model determines radiative (short- and longwave) and turbulent (latent and sensible) heat fluxes, and accounts for conductive heat exchange with the underlying medium, in order to

calculate surface temperature and melt. A multilayer subsurface model simulates temperature, density and water content, while accounting for snow compaction, water transport, refreezing, heat conduction, irreducible water storage, and runoff. To model seasonal snow in glacier-free terrain, the subsurface model has been extended with a soil routine (Westermann et al., 2011) to simulate permafrost thawing and freezing, and heat exchange within the soil and between the soil and overlying snow pack (if present), as described in Pramanik et al. (2018). Potential local impacts of (sparse) vegetation or surface roughness on the

surface energy balance in land areas are neglected.

New in the model code used in this study, with respect to the most recent model application in Pramanik et al. (2018), is the incorporation of a new percolation scheme (Marchenko et al., 2017b), as well as the implementation of an updated albedo scheme. A deep water percolation scheme, inspired by subsurface temperature measurements on the Lomonosovfonna ice cap (Marchenko et al., 2017b), has recently been implemented to mimic the effects of preferential flow pathways in snow/firn.

Additionally, we have extended the original snow age and snow depth dependent albedo scheme (Oerlemans and Knap, 1998). The original fixed characteristic time-scale for exponential decay of snow albedo due to ageing has been replaced with a temperature dependent time-scale ($t^*$). As in Bougamont et al. (2005), snow albedo decays fastest when the surface is melting ($t^*$=15 d), and for dry snow $t^*$ linearly increases from 30 to 100 days between 0 and -10 °C. The updated albedo scheme avoids overestimation of the albedo of melting surfaces in the early melt season.

The climatic mass balance refers to the sum of the surface mass balance and internal mass balance (Cogley et al., 2011) and thereby accounts for internal accumulation, i.e. refreezing and liquid water storage below the previous summer surface. Here it is calculated as the sum of mass fluxes at the surface, including precipitation (+) and moisture exchange (+/-), and mass loss through runoff (-) at the the snow/firn to ice transition (i.e. at the surface in absence of snow). No horizontal exchange of liquid water is accounted for, i.e. runoff is assumed to occur locally.

The simulation covers the period from 1 September 1957 to 31 August 2018 with a 3-hourly temporal resolution on a distributed 1-km resolution grid. We initialize the simulation by performing a 25-year spin-up using input data for the period 1957−1982, to generate initialized subsurface conditions. The subsurface model uses a Lagrangian grid to avoid numerical diffusion; surface mass fluxes due to precipitation, melt and moisture exchange induce thickness changes in the uppermost model layer with a thickness between 0 and 0.1 m. For both glacier-covered and land grid cells, a vertical grid consisting of

50 vertical layers is used. On glaciers, layer thickness doubles at the 15th, 25th and 35th layer through layer merging/splitting to yield vertical layer thicknesses from <0.1 to 0.8 m down to a depth of up to 20 m below the surface. In land areas, a fixed (initial) layer thickness of 0.1 m is used, extending to a depth of up to 5 m below the surface. Snow layer thickness gradually decreases over time due to snow compaction, which results in a lower total depth for grid cells with deep snow/firn columns. A central differencing scheme is used to simulate heat conduction, in which adaptive time-stepping assures stability; a zero heat

flux is assumed at the lower boundary (Van Pelt and Kohler, 2015).




## 3.2 Calibration

Extensive calibration of energy balance model parameters in applications on Svalbard has previously been described in Van Pelt et al. (2012) and Van Pelt and Kohler (2015). Here, we use the parameter setup as described in Van Pelt and Kohler (2015), and only recalibrate constants to which melt rates have previously been found to be most sensitive, including the background

turbulent exchange coefficient ($C_b$), the snow-to-rain transition temperature ($T_{sr}$), the fresh snow albedo ($\alpha_{fs}$), and the snowfall threshold at which the albedo is reset to the fresh snow albedo ($P_{th}$). Additionally, since the simulated climatic mass balance is highly sensitive to the downscaling of precipitation from the regional climate model grid onto the 1×1 km model grid, we also calibrate the precipitation downscaling function.

   In the first calibration step, multi-year records of $SW_{net}$ observations from KNG, HDF and NBR (Table 1) were used to

collectively calibrate $\alpha_{fs}$ and $P_{th}$. Since we aim to calibrate only fresh snow albedo and minimum snowfall to reset to the fresh snow albedo, we have selected $SW_{net}$ measurements for the period April to June, when melt effects on albedo are limited, but solar insolation is high. A two-parameter exploration revealed a lowest average root-mean-square error (RMSE) between modelled and observed daily $SW_{net}$ of 14.9 W m$^{-2}$ for the three glaciers when choosing $\alpha_{fs} = 0.83$ and $P_{th} = 0.1$ mm w.e. hr$^{-1}$. RMSE values ranged from minimum 14.1 W m$^{-2}$ on NBR to maximum 15.6 W m$^{-2}$ on HDF, suggesting consistent

performance on the three glaciers.

   In the second calibration step, stake winter balance data from eight glaciers in Svalbard (Fig. 1; Table 1) were used to calibrate coefficients in the function used to project precipitation from the coarser regional climate model grid onto the finer model grid. The function describes the distribution of precipitation accounting for local topography not captured by the regional climate model, and is formulated as an elevation-dependent relation following Van Pelt et al. (2016a):

$$Pr = Pr_0 \left[ K_1 + (z - z_0) K_2 \right] \qquad (1)$$

where $Pr$ is corrected precipitation, $z$ is elevation, $Pr_0$ and $z_0$ are spatially interpolated precipitation and elevation from the regional climate model grid onto the 1-km grid, and $K_1$ and $K_2$ are calibration coefficients. While $K_1$ is used to correct for potential biases in the regional climate model precipitation, $K_2$ represents the local precipitation-elevation gradient, which, since it is a fractional (or relative) coefficient, generates steeper absolute precipitation-elevation gradients in regions with higher

overall precipitation amounts. Using a total of 1438 stake winter balance measurements between 1967−2015, we performed a two-parameter search to find optimum values for $K_1$ (1.11) and $K_2$ (0.0022 m$^{-1}$) for which a minimum RMSE of 0.23 m w.e. was found between modelled and observed winter balance (Fig. 3, Table 2). These values imply that we apply an 11% bias correction to the regional climate model precipitation and a local precipitation lapse rate of 22% per 100 m to correct for the orographic effect due to local topography. Precipitation is set to not increase further above 900 m a.s.l., in line with an

observed negligible elevation gradient of $b_w$ above this elevation on Lomonosovfonna, central Svalbard (Van Pelt et al., 2014) and Holtedahlfonna, western Svalbard (Van Pelt and Kohler, 2015). Elevations on the 1×1-km grid do not exceed 900 m a.s.l. in southern and northeastern Svalbard.

   The final calibration step uses the stake summer balance data to optimize two parameters ($C_b$ and $T_{sr}$) that have a strong impact on summer melt. A two-parameter exploration revealed a minimum RMSE of 0.34 m w.e. between modelled and





**Table 2.** Comparison of simulated and observed $b_w$, $b_s$ and $b_n$ after calibration.

|          | Bias (m w.e.) | | | RMSE (m w.e.) | | |
|----------|---------|---------|---------|---------|---------|---------|
|          | $b_w$ | $b_s$ | $b_n$ | $b_w$ | $b_s$ | $b_n$ |
| BRG      | +0.01  | +0.08  | +0.08  | 0.14 | 0.35 | 0.39 |
| MLB      | −0.04  | +0.06  | +0.02  | 0.12 | 0.34 | 0.36 |
| KNG      | −0.12  | −0.10  | −0.21  | 0.20 | 0.30 | 0.37 |
| HBR      | −0.15  | −0.01  | −0.16  | 0.31 | 0.41 | 0.54 |
| HDF      | +0.07  | +0.07  | +0.14  | 0.14 | 0.26 | 0.30 |
| LNB      | −0.11  | +0.31  | +0.21  | 0.22 | 0.50 | 0.54 |
| AUS      | +0.20  | +0.02  | +0.23  | 0.30 | 0.29 | 0.43 |
| NBR      | +0.28  | +0.14  | +0.41  | 0.33 | 0.40 | 0.65 |
| All data | **−0.00** | **+0.03** | **+0.02** | **0.23** | **0.34** | **0.43** |

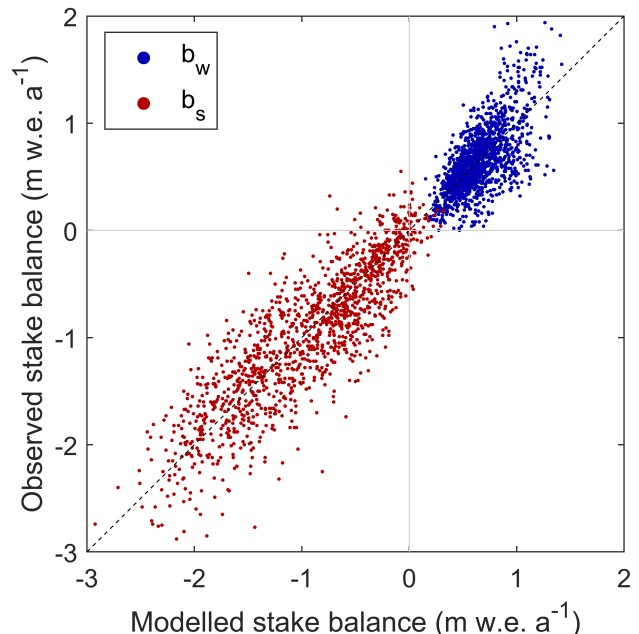

**Figure 3.** Simulated vs. observed summer and winter balance for all available stake data from eight glaciers (Table 1).

observed summer balance for a total of 1341 observations between 1967−2015 (Fig. 3) when choosing values for $C_b = 0.0025$ and $T_{sr} = 0.6\,°C$.

Altogether, comparing modelled and observed net mass balance reveals an RMSE of 0.43 m w.e for all data (Table 2). For comparison, Østby et al. (2017) previously reported an RMSE of 0.59 m w.e. using a similar set of stake data for calibration.





Contributing errors to the net mass balance RMSE include uncertainty in stake readings and bulk density estimation, model physics, climate forcing, and uncertainty in comparing observed point-values with simulated spatially-averaged values − the latter is particularly significant for locations where wind has a major impact on the snow distribution (e.g., Van Pelt et al., 2014). After calibration, remaining biases (modelled minus observed) of the winter, summer and net balance are −0.00, 0.03

and 0.02 m w.e. respectively for all data, which implies low systematic errors for long-term area-averaged climatic mass balance. Comparing net mass balance for individual glaciers reveals biases ranging from −0.21 m w.e. (KNG) to +0.41 m w.e. (NBR), while RMSE is found to range from 0.30 m w.e. on HDF to 0.65 m w.e. on NBR (Table 2). Overall, we find largest errors for NBR in central Svalbard, which is primarily caused by a substantial overestimation of $b_w$, which in turn also induces an overestimation of $b_s$ (underestimation of summer melt) due to a snow - albedo feedback. It is known that snow

accumulation on NBR is highly influenced by wind driven snow redistribution and erosion (Van Pelt et al., 2014). This may explain the overestimation of snow accumulation in our modelling of NBR, since effects of wind on snow accumulation are not accounted for in the downscaling of regional climate model precipitation. On the other hand, underestimation of $b_w$ is apparent for KNG and HBR (Fig. 3, Table 2), which results from underestimated orographic precipitation at high elevations on these glaciers. In general, the strong connection between uncertainty in $b_w$ and $b_n$ emphasises the need for precipitation calibration

for accurate CMB modelling.

### 3.3  Validation

To assess how well the model is able to simulate time-evolution of glacier-wide CMB, we compare simulated glacier-average winter CMB ($B_w$), summer CMB ($B_s$) and net CMB ($B_n$) for HBR and KNG against observation-based estimates from the WGMS database (Fig. 4). The long-term WGMS records in Svalbard from BRG and MLB are excluded due to a lack of model

grid cells falling within the glacier outlines (9 for BRG and 5 for MLB); model grids of HBR and KNG include 66 and 110 grid cells respectively. Simulated annual net CMB values show good agreement with the WGMS values for both KNG ($R = 0.86$; $P < 0.001$; RMSE$= 0.18$ m w.e. a$^{-1}$) and HBR ($R = 0.67$; $P < 0.001$; RMSE$= 0.27$ m w.e. a$^{-1}$). Furthermore, long-term simulated and observed net CMB trends are consistent for both KNG (modelled $-0.18\pm0.11$ m w.e. a$^{-1}$ decade$^{-1}$; observed $-0.10\pm0.13$ m w.e. a$^{-1}$ decade$^{-1}$) and HBR (modelled $0.02\pm0.20$ m w.e. a$^{-1}$ decade$^{-1}$; observed $-0.05\pm0.19$ m w.e. a$^{-1}$

decade$^{-1}$).

Air temperature and precipitation are the main meteorological drivers of spatial patterns and trends in CMB and its components. As discussed in Sect. 3.1, the downscaling of precipitation has been optimized using in situ winter balance data from multiple sites in Svalbard. Here, we validate the temperature forcing by comparing downscaled daily 2-m temperature with in situ temperature records (recorded at $1-4$ m heights) from five sites in Svalbard (Table 1; Sect. 2.3). Results are summarized

in Table 3. We find very high correlations ($R =0.95-0.97$; $P < 0.001$), RMSE ranging between 2.0 °C (KNG) and 4.6 °C (HDF), and biases ranging from $-2.3$ °C (AUS) to $+0.7$ °C (KNG). In general, we find good agreement between downscaled and observed temperatures both glacier and non-glacier terrain in different regions in Svalbard. The largest bias and RMSE are found at AUS in northeast Svalbard, which can be ascribed to a substantial underestimation of air temperature during the cold season (September−May) of $-3.2$ °C, whereas the summer (June−August) air temperature bias is small (+0.4 °C).





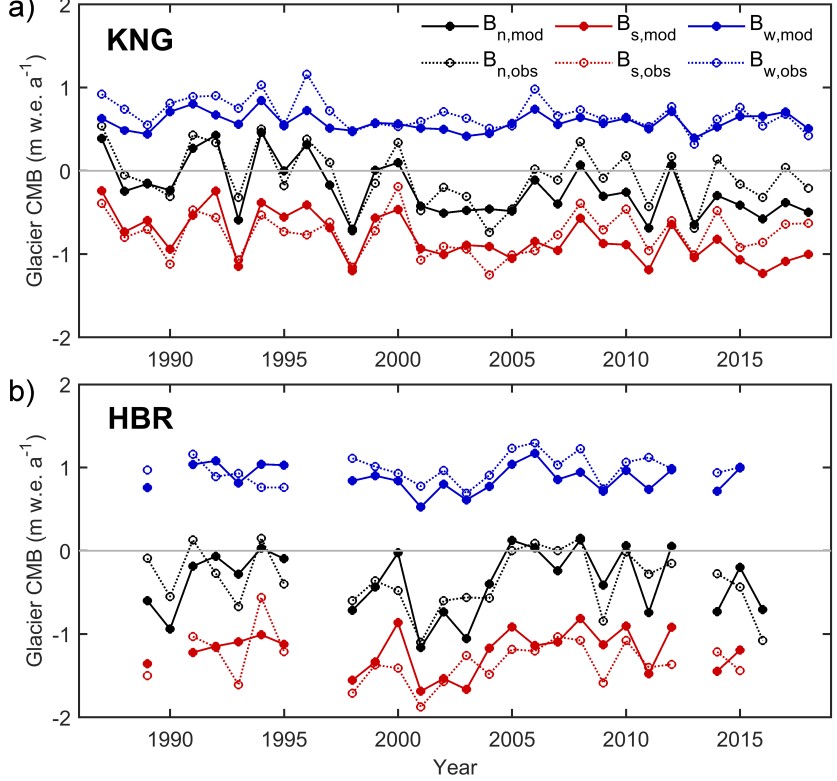

**Figure 4.** Comparison of simulated glacier-wide summer, winter and net mass balance against WGMS records for KNG (a) and HBR (b).

**Table 3.** Comparison of downscaled and observed air temperatures at glacier- and land-based weather stations.

| Location | Elevation | Surface type | # of observ. | $R$ | bias | RMSE |
|---|---|---|---|---|---|---|
| | (m a.s.l.) | | (days) | | (°C) | (°C) |
| LYB | 28 | land | 14963 | 0.97 | +0.4 | 2.6 |
| NA | 8 | land | 17066 | 0.96 | −0.1 | 2.3 |
| KNG | 520 | glacier | 1374 | 0.97 | +0.7 | 2.0 |
| HDF | 680 | glacier | 1334 | 0.95 | +0.1 | 3.1 |
| NBR | 519 | glacier | 1554 | 0.95 | −0.8 | 2.9 |
| AUS | 350 | glacier | 4386 | 0.92 | −2.3 | 4.6 |

Finally, in situ observations from shallow cores (Sect. 2.3) are used to validate bulk density ($\rho_{sub}$) simulated at AUS, HDF, NBR and KNG during four years down to depths of $7-15$ m (Table 1). For three sites, we find negative model biases for $\rho_{sub}$ of $-25$ kg m$^{-3}$ (NBR), $-30$ kg m$^{-3}$ (AUS) and $-38$ kg m$^{-3}$ (HDF). On KNG, a positive bias of $+48$ kg m$^{-3}$ is found. Table 2





shows that KNG is the only site of the four experiencing a negative $b_w$ bias. Based on this, we argue that an underestimation of accumulation explains the overestimation of $\rho_{sub}$ at KNG, and vice versa at NBR, AUS and HDF. An inverse relation between $\rho_{sub}$ and accumulation follows from 1) the parametrization used for gravitational settling (Ligtenberg et al., 2011), and 2) an increased significance of refreezing on the vertical density distribution where accumulation rates are low (subsurface layers

remain closer to the surface for a longer time and will hence experience refreezing of stored water in the cold season during more years).

## 4   Results & Discussion

In this section, we present and discuss spatial patterns and trends of simulated CMB, ELA, subsurface conditions, refreezing and runoff over the period 1957−2018.

### 4.1   Climatic mass balance & ELA

Averaged over the entire simulation period, we find a spatial mean glacier CMB of +0.09 m w.e. a$^{-1}$, which is comparable to Østby et al. (2017) [+0.08 m w.e. a$^{-1}$ over the period 1957−2014] and more positive than a recent estimate by Möller and Kohler (2018) [−0.03 m w.e. a$^{-1}$ over the period 1957−2010]. The spatial CMB distribution in Fig. 5a reveals most negative CMB values (down to −2.5 m w.e. a$^{-1}$) at low elevations in southern and western Svalbard, and most positive CMB (up to

1.3 m w.e. a$^{-1}$) at high-elevation sites on the Lomonosovfonna ice cap (central Svalbard). Assuming a frontal ablation rate equivalent to −0.18 m w.e. a$^{-1}$ (Blaszczyk et al., 2009), and negligible basal melting, we estimate a total mass balance of −0.09 m w.e. a$^{-1}$. In the latter calculation it is implicitly assumed that frontal ablation rates from Blaszczyk et al. (2009) for the early 2000s apply during the whole simulation period. We find significantly negative CMB trends in southern and central Svalbard, while trends are not significant in the north (Fig. 5b). On average, a significantly negative CMB trend of −0.06±0.04

m w.e. a$^{-1}$ decade$^{-1}$ is found (Fig. 5c). For comparison, a more negative trend of −0.14 m w.e. a$^{-1}$ decade$^{-1}$ was reported by Østby et al. (2017) over the period 1957−2014, although it was argued that the trend may have been overestimated based on a comparison of long-term CMB at a single stake site on MLB. Conversely, Lang et al. (2015) found a weaker negative CMB trend (−0.03 m w.e. a$^{-1}$ decade$^{-1}$) for 1979−2013, which is however not significantly different from our trend of −0.07±0.08 m w.e. decade$^{-1}$ over the same period. Significant trends of opposite sign are found for the winter balance (+0.02±0.01 m

w.e. a$^{-1}$ decade$^{-1}$) and summer balance (−0.08±0.03 m w.e. a$^{-1}$ decade$^{-1}$), suggesting that a winter accumulation increase compensates for some of the increased summer ablation. Inter-annual variability of net CMB correlates strongly with both summer (June−August) air temperature ($R = 0.78$; $P < 0.001$) and annual (September−August) precipitation ($R = 0.60$; $P < 0.001$), while no significant correlation exists between annual temperature and net CMB ($R = 0.10$; $P > 0.1$).

    The average ELA of the entire glacierized area in Svalbard is 367 m a.s.l. for 1957−2018. The ELA distribution resembles

an earlier observation-based map by Hagen et al. (2003) with highest ELA (>700 m a.s.l.) in relatively dry regions in northern Spitsbergen and lowest ELA (<200 m a.s.l.) induced by cold conditions in northeastern Svalbard (Fig. 6a). Significant positive ELA trends are apparent for all of Svalbard, except for the most northern parts (Fig. 6b), where increased precipitation (Fig.





**Figure 5.** Long-term mean spatial CMB distribution (a) and trends (b). In (c) time-series of area-averaged annual mean summer, winter and net CMB (solid lines) and linear trends (dashed lines) are shown. In (c) years are defined based on a mass balance year between 1 September (preceding year) and 31 August.

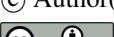



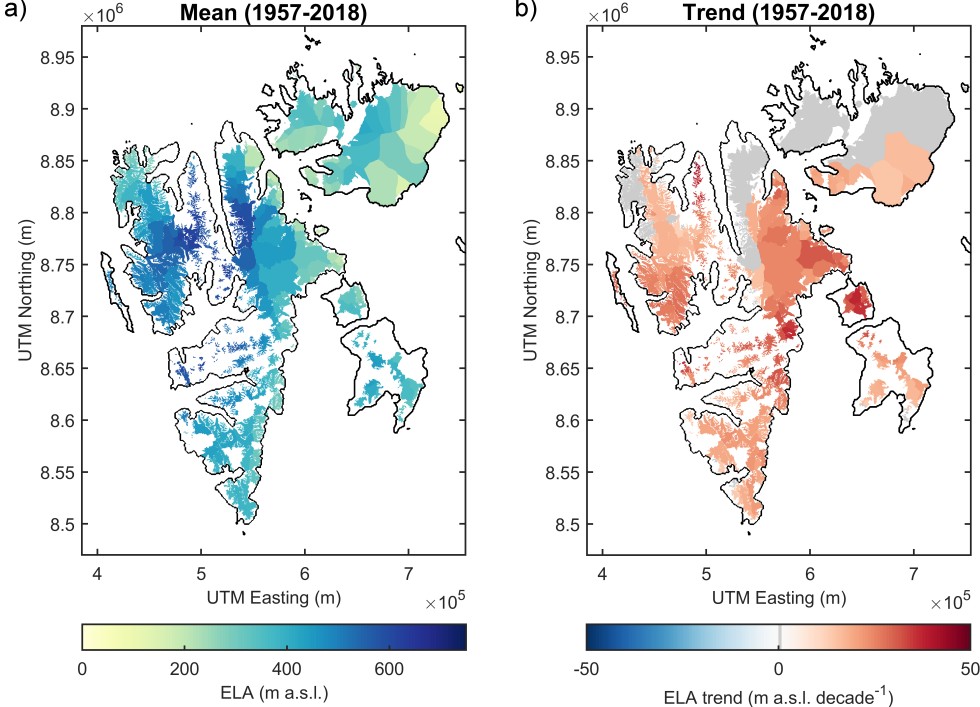

**Figure 6.** Long-term mean spatial ELA distribution (a) and trends (b). Data are averaged per glacier basin, based on the glacier outline database in König et al. (2014).

2d) offsets an ELA increase due to a melt increase. Based on annual ELA time-series, we find a significant mean positive ELA trend of $17\pm12$ m a.s.l. decade$^{-1}$, which is slightly less than a previously reported trend of 25 m a.s.l. decade$^{-1}$ over $1961-2012$ in Van Pelt et al. (2016a). As a result of upward ELA migration, the accumulation area ratio (AAR) has decreased at an absolute rate of $-4\%$ per decade$^{-1}$; the average AAR for $1957-2018$ equals 65% with annual values ranging from 17% (1997−1998) to 91% (1964−1965). As previously discussed in Van Pelt and Kohler (2015), surface melt is amplified due to substantial lowering of the albedo in the new ablation areas exposed by the retreating ELA. The average albedo over the simulation period is 0.76 for all glaciers in Svalbard, with a significant negative trend of $-0.004\pm0.001$ decade$^{-1}$ (locally down to $-0.024$ decade$^{-1}$), inducing an average 2% decade$^{-1}$ increase of absorbed solar radiation.

## 4.2 Glacier subsurface conditions

As a collective measure of density and depth of snow and firn in glacierized areas, we quantify the total pore space down to a depth of 14 m below the surface ($P_{14}$), expressed in m$^3$ m$^{-2}$, and shown in Fig. 7a-b. Large accumulation zones with $P_{14}$ exceeding 5 m$^3$ m$^{-2}$ are found at high elevations on the three major ice caps in northern Svalbard (Holtedahlfonna, Lomonosovfonna and Austfonna); smaller accumulation zones with generally lower $P_{14}$ prevail in southern Svalbard (Fig. 7a). Trends in $P_{14}$ (Fig. 7b) are most negative (down to $-0.6$ m$^3$ m$^{-2}$ decade$^{-1}$) in elevation bands close to the long-term





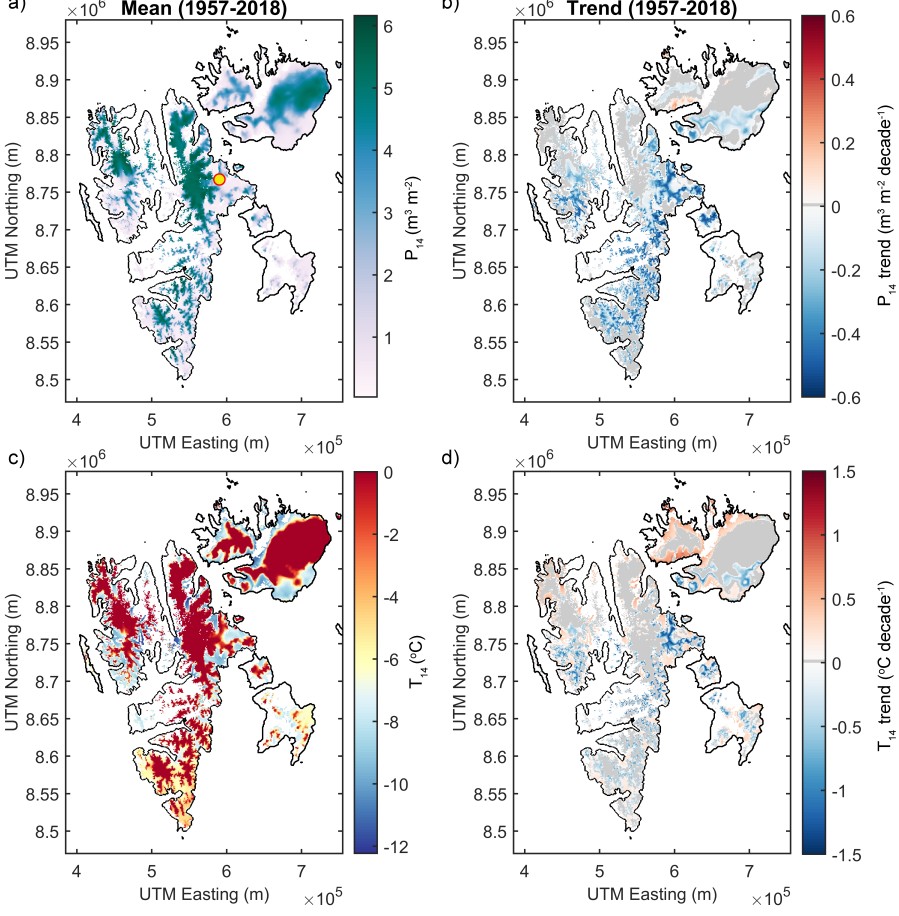

**Figure 7.** Long-term mean $P_{14}$ distribution (a) and trends (b). Long-term mean $T_{14}$ distribution (c) and trends (d). The corresponding location of the subsurface profiles in Fig. 8 is marked with a red circle in (a).

mean ELA, as upward migration of the firn line causes a major decline in firn depth. As a result, the most negative $P_{14}$ trends are found in central Svalbard, where ELA trends are most positive (Fig. 6b). For $1957-2018$, average $P_{14}$ for the glacierized area, i.e. including both ablation and accumulation zones, equals 2.3 m$^3$ m$^{-2}$; the average trend is significantly negative ($-0.09\pm0.03$ m$^3$ m$^{-2}$ decade$^{-1}$), and equivalent to a 4% decrease of $P_{14}$ per decade.

5    The distribution and trends of deep temperature ($T_{14}$), defined here as the temperature at 14 m below the surface, are shown in Fig. 7c-d for the glacierized area of Svalbard. The $T_{14}$ distribution reveals a marked transition around the ELA from cold (non-temperate) conditions in the ablation zones to temperate conditions in accumulation areas for all glaciers in Svalbard. This thermodynamic structure is common for Svalbard glaciers (Björnsson et al., 1996; Pettersson, 2004), and has previously been linked to the high significance of (deep) percolation and refreezing in accumulation zones (e.g., Van Pelt et al.,
10  2012, 2016b). Temperate $T_{14}$ conditions also precondition the potential formation of perennial firn aquifers, which have been





detected using ground-penetrating radar on Holtedahlfonna in western Svalbard (Christianson et al., 2015), and recently also on Lomonosovfonna in central Svalbard (R. Pettersson, unpublished data). The widespread occurrence of temperate deep firn suggests the likelihood of perennial firn aquifers in other accumulation zones on Svalbard. On Austfonna, a radar survey in 2014 showed a strong reflector over large distances across the summit area, which potentially results from deep slush water

storage (T. Dunse, unpublished data). In addition to temperature, other factors affecting firn aquifer development include surface topography (steering water flow in the aquifer), and the potential for englacial drainage through cracks, crevasses and moulins. Our results suggest that even the highest (coldest) accumulation zones in Svalbard have average temperate deep firn conditions. This is in line with recent measurements (2012−2015) on Lomonosovfonna at 1200 m a.s.l. (Marchenko et al., 2017b), but does not agree with earlier findings of sub-temperate conditions at ice core drill sites on Lomonosovfonna in 1997

(Van de Wal et al., 2002) and Holtedahlfonna in 2005 (Beaudon et al., 2013). However, we infer that both these drill sites were likely drilled in locations with isolated cold deep temperature conditions within otherwise temperate accumulation zones, as confirmed by the widespread presence of perennial firn aquifers. Cold deep temperature conditions may occur locally at wind-exposed sites, e.g. on an ice divide or ridge, as accumulation rates are typically lower due to wind erosion, which has a cooling effect on deep firn (Kuipers Munneke et al., 2014). Additionally, we infer that the convex topography of ice divides promotes

efficient drainage and reduces the significance of latent heat release by refreezing. For both drill sites, reported accumulation rates estimated from the ice cores of 0.41 m w.e. a$^{-1}$ (Lomonosovfonna, 1950−1997, Pohjola et al., 2002) and 0.50 m w.e. a$^{-1}$ (Holtedahlfonna, 1963−2005, Van der Wel et al., 2011) are indeed substantially lower than observed at the nearest stakes on Holtedahlfonna (0.98 m w.e. a$^{-1}$ for 2003−2015) and Lomonosovfonna (0.85 m w.e. a$^{-1}$ for 2006−2015). Long-term trends of $T_{14}$ (Fig. 7d) reveal a warming trend in ablation zones and a cooling trend in former accumulation zones that recently became

ablation zone due to upward migration of the firn line; the average trend is weakly negative ($-0.03\pm0.03$ $^{o}$C decade$^{-1}$).

An example of firn density and temperature evolution during two periods (1964−1968 and 2014−2018) at a site close to the long-term mean ELA in central Svalbard is shown in Figure 8 (location indicated in Fig. 7). During 1964−1968, deep temperature is consistently at the melting point (Fig. 8c) and no thick ice layers are present in the upper 10−15 m of firn (Fig. 8a). During 2014−2018, the same site is still in the lower accumulation zone, but now firn density is markedly increased, with

impermeable ice below a depth of 1−3 m below the surface (Fig. 8b). It is noteworthy that similar firn developments have recently been observed in the lower accumulation zone in western Greenland (Cox et al., 2015; Machguth et al., 2016) and the Canadian Arctic (Bezeau et al., 2013), and have been argued to potentially affect horizontal drainage. As firn densifies, percolating water more readily runs off, and the potential for deep water storage and subsequent refreezing is reduced. In response to reduced refreezing, as well as faster heat conduction, deep firn/ice temperatures in 2014−2018 are no longer

temperate at this site (Fig. 8d).

## 4.3  Refreezing

The distribution of refreezing for both glacier-covered and land areas reveals that the highest refreezing rates (up to 0.41 m w.e. a$^{-1}$) are in the accumulation zones (Fig. 9a), where percolating water can be stored deep in snow/firn and refreeze over the course of the winter season (Van Pelt et al., 2016b). The lowest refreezing rates (<0.05 m w.e. a$^{-1}$) are at low elevations, i.e. in





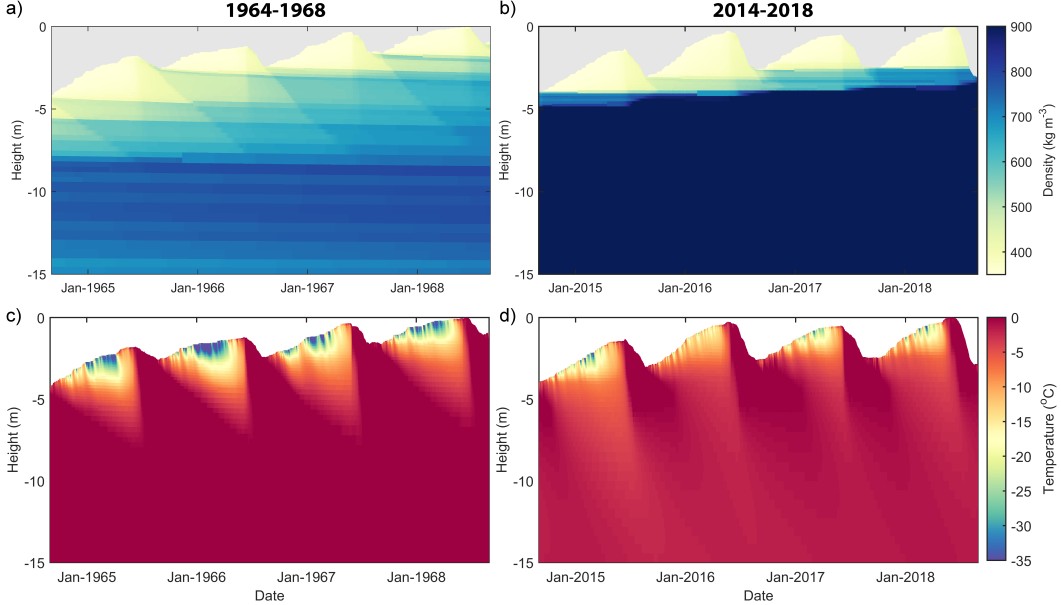

**Figure 8.** Subsurface density (a-b) and temperature (c-d) evolution during the periods 1964−1968 (a&c) and 2014−2018 (b&d). The corresponding geographic location of the site is indicated in Fig. 7a.

coastal regions and valleys, where thin seasonal snow packs develop over winter, thereby limiting the potential for refreezing. For 1957−2018, we find average refreezing rates of 0.24 m w.e. a$^{-1}$ and 0.14 m w.e. a$^{-1}$ for the glacier-covered and land areas respectively. For comparison, Østby et al. (2017) found comparable refreezing rates of 0.22 m w.e. a$^{-1}$ for all glaciers in Svalbard during 1957−2014. On average, we find that 25% of melt and rainwater is refrozen, implying a substantial reduction

of runoff. It should however be acknowledged that indirect effects after refreezing, in particular heat release in the snow pack, will induce additional melt, which will reduce the net impact of refreezing on runoff (Van Pelt et al., 2016b). Long-term trends reveal significantly decreasing refreezing rates (down to −0.03 m w.e. decade$^{-1}$) primarily at elevations around the ELA in response to firn line retreat (Fig. 9b). No significant trends in refreezing are found in high accumulation zones, which implies the likely growth of perennial firn aquifers during the simulation period since input from surface melt and rainfall shows a

clear positive trend (+0.058±0.022 m w.e. decade$^{-1}$). On average, we find comparable negative trends for the glacier-covered areas (−0.007±0.002 m w.e. a$^{-1}$ decade$^{-1}$) and land areas (−0.008±0.002 m w.e. a$^{-1}$ decade$^{-1}$), implying a much faster *relative* decrease of refreezing on land (−6.0% decade$^{-1}$) than on glaciers (−2.9% decade$^{-1}$). As deep cold firn has been absent throughout the simulation period, there is no potential for additional refreezing buffering higher melt rates in a warming climate, which is similar to what has been suggested for peripheral glaciers and ice caps of the Greenland ice sheet beyond a

'tipping point' in 1997 (Noël et al., 2017). The consistently negative refreezing trend throughout the simulation period in this study suggests that the tipping point would have occurred already prior to the start of the simulation in 1957. Similar long-term negative refreezing trends were previously described by Noël et al. (2018) for ice caps in the southern Canadian Arctic. Future





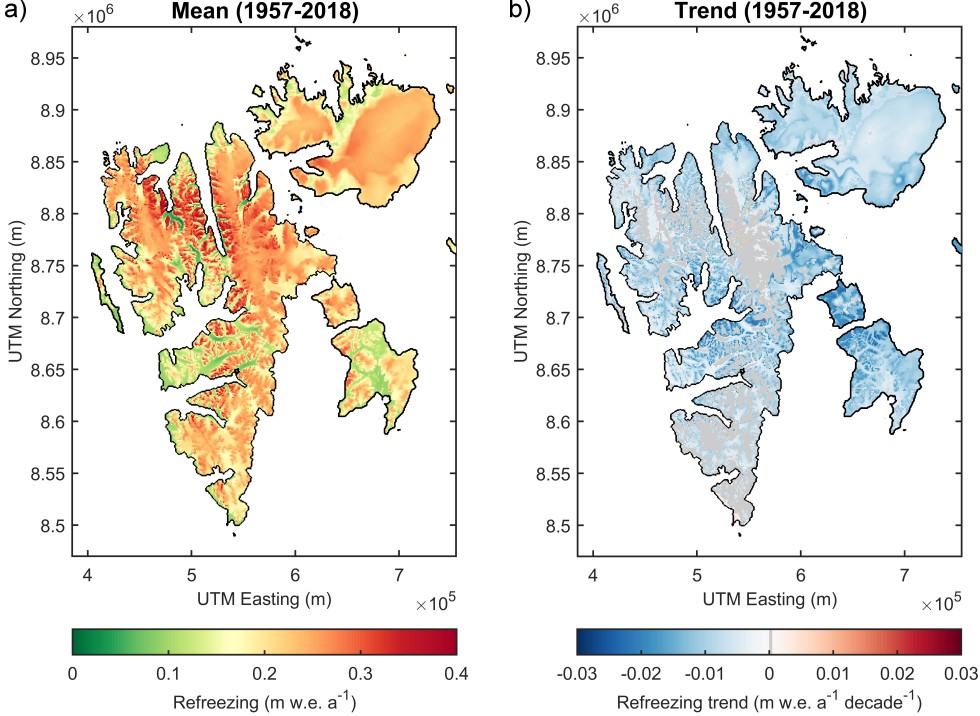

**Figure 9.** Long-term mean spatial refreezing distribution (a) and trends (b).

projections of refreezing in Svalbard show that while there will be less refreezing in the early melt season due to reduced winter cooling (reducing the cold content required for refreezing) and shrinking accumulation zones, at the same time winter-time refreezing during and after rainfall and melt events will increase (Van Pelt et al., 2016b).

## 4.4 Seasonal snow cover duration

Land areas and glacier ablation zones in Svalbard experience snow-free conditions during the summer season. The extent of the snow-free season is defined by the snow disappearance date, which we define to occur when snow amount first drops below a threshold (1 cm w.e.), and the snow onset date, which we define as the first date on which snow (>1 cm w.e.) accumulates and remains until next year. Long-term mean distributions of the snow disappearance and onset dates (Fig. 10a and c) show that the earliest snow disappearance (late May) and latest snow arrival (late October) are to be found in the relatively dry

valleys of central Svalbard. Trends in the snow disappearance date are primarily controlled by winter accumulation (cumulative snowfall) and melting. We find negligible trends of the snow disappearance date for most of Svalbard, except for parts of central Svalbard, where snow disappears earlier over time (up to 4 days decade$^{-1}$, Fig. 10b). There is however no significant average snow disappearance trend for all of Svalbard ($0.0 \pm 0.9$ days decade$^{-1}$), suggesting that, on average, the slight increase in precipitation, generating thicker winter snow packs, is compensated for by an earlier onset of melting. The snow onset date

(September−October) is strongly influenced by air temperature affecting both precipitation type (snow/rain) and potential melt





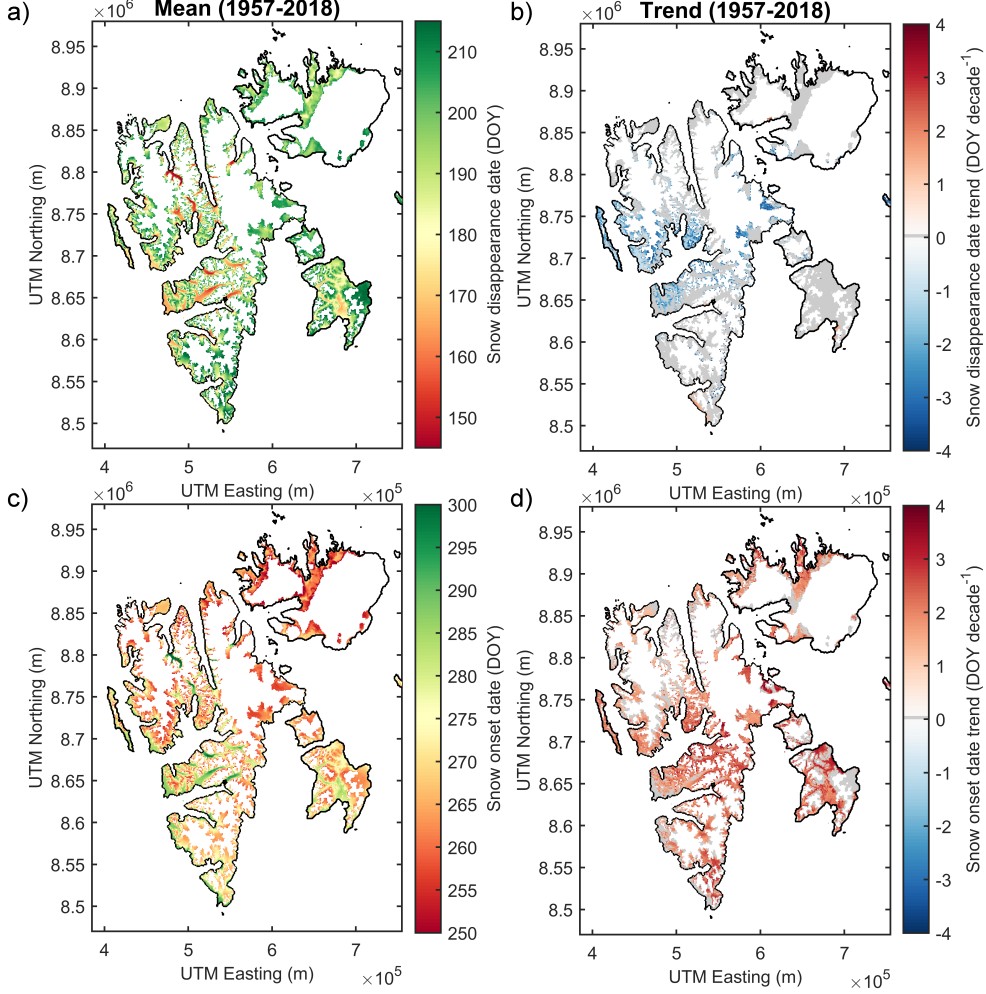

**Figure 10.** Long-term mean spatial snow disappearance date distribution (a) and trends (b). Long-term mean spatial mean snow onset date distribution (c) and trends (d). Snow onset and disappearance dates are only calculated for sites where snow melts completely in summer during at least half of the years in the simulation.

of freshly fallen snow. In response to the substantial autumn warming (Førland et al., 2011; Van Pelt et al., 2016a), snow onset trends are significantly positive (up to $+4$ days decade$^{-1}$) for most of Svalbard (Fig. 10d), leading to a significant mean positive snow onset date trend of $+1.4 \pm 0.9$ days decade$^{-1}$. For comparison, Van Pelt et al. (2016a) found comparable trends of $+1.8$ days decade$^{-1}$ for the snow onset date and $+0.7$ days decade$^{-1}$ for the snow disappearance date over the shorter period 1957−2012.





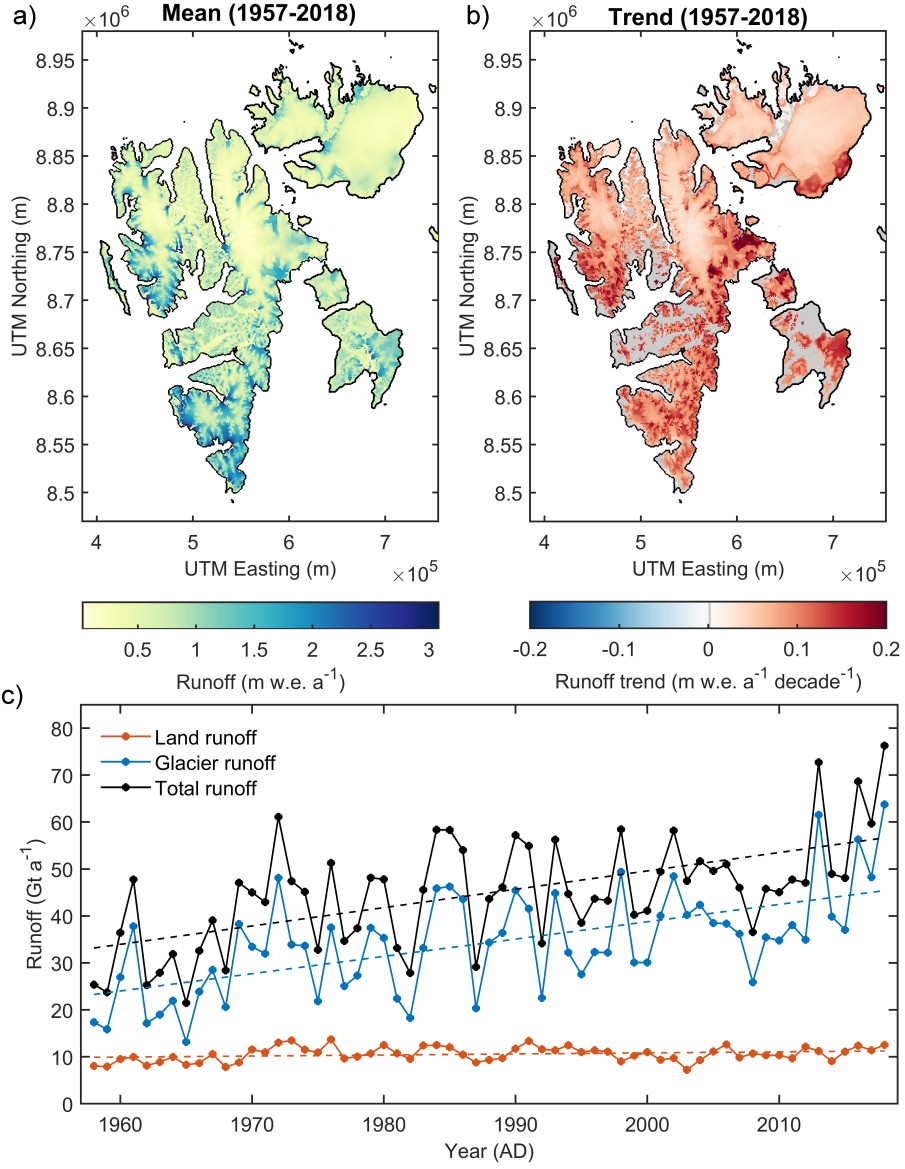

**Figure 11.** Long-term mean spatial runoff distribution (a) and trends (b). In (c) time-series of area-averaged annual glacier, land and total runoff (solid lines) and linear trends (dashed lines) are shown. Years in (c) are defined between 1 September (preceding year) and 31 August.

## 4.5 Runoff

The long-term mean runoff distribution (Fig. 11a) shows local discharge is apparent across all of Svalbard, with the highest rates ($> 3$ m w.e. a$^{-1}$) in the glacier ablation zones in southern Svalbard, and the lowest rates $< 0.3$ m w.e. a$^{-1}$ at the high elevations of the Lomonosovfonna ice cap in central Svalbard. Melt rates on land are limited to the amount of seasonal snow accumulating





during the cold season, and therefore generate much lower runoff rates than nearby glacier sites at similar elevations (Fig. 11a). As a result, the area-averaged runoff from glaciers (0.81 m w.e. a$^{-1}$) is higher than the runoff from land (0.63 m w.e. a$^{-1}$), despite the lower mean elevation of the land cells compared to the glacier grid. Trends of runoff over the simulation period (Fig. 11b) are generally not significant for land, but are significantly positive for glaciers, with largest increases (up to 0.2

m w.e. a$^{-1}$ decade$^{-1}$) in ablation zones recently exposed by the retreating ELA. Time-series of runoff in Gt a$^{-1}$ (Fig. 11c) show average runoff of 10.6 and 34.3 Gt a$^{-1}$ from land and glacier-covered areas respectively, contributing to a total average annual runoff of 44.9 Gt a$^{-1}$. Runoff from land is primarily controlled by precipitation, and as a result the long-term trend is not significant ($+0.2 \pm 0.3$ Gt a$^{-1}$ decade$^{-1}$). Conversely, runoff from glaciers is primarily controlled by summer melt, and is found to increase markedly over the simulation period ($+3.7 \pm 1.3$ Gt a$^{-1}$ decade$^{-1}$), in accordance with decreasing

CMB. As a result, total runoff increases by $+3.9 \pm 1.4$ Gt a$^{-1}$ decade$^{-1}$, which is equivalent to a 9% decade$^{-1}$ increase in runoff. The contrast in trends of runoff from glaciers and land implies a substantial decrease in the relative contribution of land runoff to total runoff from ~30% to around ~20% between 1957 and 2018. Finally, the Svalbard averaged trend in runoff ($+0.065 \pm 0.023$ m w.e. decade$^{-1}$) is substantially larger than the trend in the sum of melt and rainfall ($+0.058 \pm 0.022$ m w.e. decade$^{-1}$), which is fully explained by a negative trend in refreezing ($-0.007 \pm 0.002$ m w.e. decade$^{-1}$). That means that 11%

of the increase in runoff can be explained by reduced refreezing over the simulation period.

## 4.6   Uncertainties

As described in Sect. 3.2 and in previous studies using the same model in Svalbard (Van Pelt et al., 2012; Van Pelt and Kohler, 2015; Marchenko et al., 2017b; Pramanik et al., 2018), observational data have been extensively used for calibration, thereby reducing errors in downscaling climate input, solving the energy balance and simulating subsurface conditions. Nevertheless,

uncertainty remains, and here we briefly summarise the main remaining sources of errors.

First, we assumed the elevation grid and glacier masks to be fixed throughout the simulation period. As both elevations and masks are based on observational data collected after 1990, this may introduce biases in particular during the first decades of the simulation. The uncertainty due to elevation errors is most pronounced near glacier fronts where thinning rates between $1-2$ m a$^{-1}$ have been observed in the $2-4$ decades preceding $2003-2007$ (Nuth et al., 2010). With a mean balance gradient

of 0.002 m w.e. a$^{-1}$ m$^{-1}$, this would generate a potential underestimation of CMB of $0.1-0.2$ m w.e. a$^{-1}$ during the first years of the simulation at sites near the glacier snout; at higher elevations errors will be markedly smaller. Typical errors associated with the use of a fixed glacier mask, compared to a time-dependent glacier mask, have previously been quantified for Svalbard for a similar simulation period at around $0.02-0.04$ m w.e. a$^{-1}$ (Østby et al., 2017). We assume similar errors would apply here. It is noteworthy that CMB errors induced by a fixed mask will be of opposite sign as errors induced through the use of

a fixed DEM (underestimation of glacier extent in the early decades leads to a too positive CMB, while underestimation of elevations induces a too negative CMB), meaning that some of the above errors are likely to cancel each other out. Largest errors will apply to glaciers that surged during the simulation period. The use of a fixed mask and elevations has the advantage that all presented trends in climatic mass balance and related products can be attributed to changes in the climate forcing, and we can exclude any influences from dynamically induced geometric changes.



A second source of error comes from uncertainty in the climate input, more specifically the air temperature and precipitation forcings, to which climatic mass balance, seasonal snow development and derived products are most sensitive. Validation of air temperature against glacier- and land-based measurements (Sect. 3.3) revealed good correlation and generally low biases. In turn, winter balance data were used to optimize the downscaling of precipitation, also returning good correlation and negligible

biases (Sect. 3.2). Nevertheless, on average we find a substantially higher snowfall rate (0.89 m w.e. a$^{-1}$) than previously reported rates of 0.61 m w.e. a$^{-1}$ by Østby et al. (2017) for 1957−2014, and 0.44 m w.e. a$^{-1}$ by Lang et al. (2015) for 1979−2013. Østby et al. (2017), however, found that winter accumulation was generally underestimated, primarily at higher elevations, based on a comparison with similar stake winter balance data as used in this study (Fig. 11 in Østby et al., 2017). Furthermore, Lang et al. (2015) only validated their precipitation estimates against meteorological station data in Svalbard,

which are known to suffer severely from undercatch (Førland and Hanssen-Bauer, 2000). Nevertheless, we cannot rule out potential biases in our snowfall/precipitation estimates, in particular because all stakes used for calibration are located on glaciers and primarily along their centerlines, which may induce potential biases (e.g., Nuth et al., 2012; Deschamps-Berger et al., 2019). Additionally, the relatively coarse spatial resolution of the regional climate model forcing may cause spatial precipitation fields to miss some of the impacts of terrain on the precipitation distribution, even though this is to some extent

compensated for by the precipitation downscaling, which accounts for local elevation. Finally, as also discussed in Sect. 3.2, the inconsistency between the point-wise nature of stake observations and gridded model output representing processes within 1 km$^2$ cells, induces uncertainty in the comparison of climatic mass balance components. This is likely to be most pronounced for the $b_w$ comparison in wind-affected areas across Svalbard, since $b_w$ is known to vary over distances much smaller than the 1 km horizontal resolution used here (e.g., Winther et al., 2003; Van Pelt et al., 2014).

A third source of uncertainty are the modelling errors, which includes uncertainties related to solving the energy balance, simulating subsurface conditions as well as model initialization. Descriptions of the heat fluxes comprising the surface energy balance have been optimized against observational data in glacier basin studies on Nordenskiöldbreen (Van Pelt et al., 2012) and around Kongsfjorden (Van Pelt and Kohler, 2015), as also discussed in Sect. 3.2. With the exception of the new albedo parametrization, which we calibrated against observed $SW_\text{net}$ data, other energy balance parameters were taken as in the

aforementioned studies. As AWS data from only two sites in central and western Svalbard were used for energy balance model calibration, potential biases may arise for other areas in Svalbard. Regarding uncertainty in simulating subsurface conditions, it is worth noting that the recently implemented deep water percolation scheme (Sect. 3.1, Marchenko et al., 2017b) significantly reduces uncertainty in simulated firn temperatures compared to the earlier bucket scheme, which was found to underestimate rapid deep transport of water through piping. Furthermore, the comparison of simulated and observed bulk firn density shows

good agreement (Sect. 3.3), and suggests that model-induced biases are small. We refrain from a detailed vertical comparison of simulated firn density profiles with observed firn core data, since previous work has shown the extremely local character of firn stratigraphy in Svalbard (Marchenko et al., 2017a), due to local interactions between stratigraphy and vertical water percolation. As in previous glacier basin-scale applications, we have applied substantial spin-up (25 years) to generate subsurface conditions at the start of the simulation, which were consistent with the applied climate forcing during 1957−1982. Obviously, this

generates some uncertainty as the 1957−1982 may differ from the actual climate conditions in the decades prior to 1957.





As discussed in Van Pelt and Kohler (2015), the impacts of perturbing temperature and precipitation during initialization on simulated climatic mass balance are typically only significant in the first few years of the simulation; impacts on simulated firn air content were found to be present even after 20 years into the simulation, which is, however, likely to be less significant in this study given the relatively shallow depth of the vertical domain of <20 m.

**5   Conclusions**

We present a model dataset of climatic mass balance, snow conditions and runoff for all of Svalbard for the period $1957-2018$. Output with a 3-hourly temporal and $1 \times 1$-km spatial resolution is generated with a coupled surface energy balance $-$ snow/firn/soil model. The model is forced with downscaled regional climate model fields and applied to both glacier-covered and land areas. In situ observational data from mass balance stakes, weather stations and shallow cores are used for model calibration

and/or validation of the results. Based on the model output we analyze spatial variability and trends of climatic mass balance, equilibrium line altitude, glacier subsurface conditions, refreezing, seasonal snow season length and runoff.

We find an area-averaged positive CMB ($+0.09$ m w.e. a$^{-1}$), and a significant negative longterm trend ($-0.06$ m w.e. a$^{-1}$ decade$^{-1}$) over the simulation period. The negative CMB trend has caused the ELA to increase ($+17$ m decade$^{-1}$) and the AAR to decrease ($-0.04$ decade$^{-1}$) markedly. These trends are significant for all of Svalbard, except for the most northern regions.

Retreat of the ELA causes a significant reduction of mean firn air content ($-0.09$ m decade$^{-1}$), with the most pronounced changes (down to $-0.6$ m decade$^{-1}$) in ablation areas that were recently exposed by the retreating ELA. These new ablation zones also experience a strong decrease in temperature at 14 m depth (down to $-1.5$ $^o$C decade$^{-1}$), while the remainder of the ablation zones show a general warming trend. All high-altitude accumulation zones are found to exhibit temperate deep firn conditions, suggesting the potential for wide-spread presence of firn aquifers across Svalbard. We find average refreezing rates

of 0.24 m w.e. a$^{-1}$, showing pronounced negative trends for both glacier-covered areas ($-0.007$ m w.e. a$^{-1}$ decade$^{-1}$) and land areas ($-0.008$ m w.e. a$^{-1}$ decade$^{-1}$). Increased precipitation and melt cause the date of disappearance of seasonal snow packs to remain stable throughout the simulation period, while increased autumn temperatures induce a significant increase in the date of seasonal snow onset ($+1.4$ days decade$^{-1}$). The average total runoff for Svalbard (44.9 Gt a$^{-1}$) is dominated by runoff from glaciers (34.3 Gt a$^{-1}$) rather than runoff from land (10.6 Gt a$^{-1}$). A strong positive runoff trend applies to glacier

runoff ($+3.7$ Gt a$^{-1}$ decade$^{-1}$), while runoff from land remained nearly stable ($+0.2$ Gt a$^{-1}$ decade$^{-1}$), causing an increase in the relative contribution of glacier discharge to total runoff from 70 to 80% over the simulation period.

*Data availability.* The digital elevation model can be accessed at https://doi.org/10.21334/npolar.2014.dce53a47 (Norwegian Polar Institute, 2014). The glacier and land mask were constructed from glacier outlines, which are available at https://doi.org/10.21334/npolar.2013. 89f430f8 (König et al., 2013). The model output behind the presented figures of air temperature, precipitation, CMB, ELA, runoff, refreezing,

$T_{14}$, $P_{14}$, snow onset and disappearance dates are available in the following repository: https://doi.org/10.6084/m9.figshare.7836530.v1 (Van Pelt et al., 2019). The full model dataset, of which only a selection is presented here, contains data with a 3-hourly temporal resolution and for an extended set of variables; an overview of the readily available data can be found at http://www.wardvanpelt.com/model_output.txt.





Glacier-wide mass balances for KNG, HBR, HDF, MLB and ABB are available in the database of the World Glacier Monitoring Service (WGMS; https://wgms.ch/). Meteorological time-series for Ny-Ålesund and Longyearbyen are accessible through the eKlima portal (Norwegian Meteorological Institute; http://eklima.met.no/). The Kongsvegen AWS time-series are also accessible at https://data.npolar.no/dataset/5dc31930-0922-4483-a1df-6f48af9e371b (Kohler et al., 2017). Unrestricted access to the HIRLAM regional climate model data, point stake

mass balance data, and the remaining AWS time-series is provided upon request by contacting the institutes that collected/generated the data (see Sect. 2).

*Author contributions.* The research idea was conceptualized by WvP, who also performed the main analysis. VP, RP, SM, JK, BL, JOH, TD, CR, TVS and WvP contributed to observational data collection and/or processing. WvP developed the model with support from SM. WvP wrote the manuscript with contributions from all co-authors.

*Competing interests.* The authors declare that they have no conflict of interest.

*Acknowledgements.* This work was performed with funding support from the Swedish Research Council (VR), Stiftelsen Ymer-80, Svalbard Integrated Arctic Earth Observing System (SIOS), Uppsala University (Finn Malmgren's stipend), the Norwegian Research Council (TIGRIF project), the Polish-Norwegian Research Programme GLAERE, and the Norwegian Polar Institute (TW-ICE project). We thank the Norwegian Meteorological Institute for providing access to the HIRLAM regional climate model output. We are further grateful to all the

15 people that assisted with stake mass balance, AWS, and shallow-core measurements on glaciers in Svalbard over the years.




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
