# Peer review of "A long-term dataset of climatic mass balance, snow conditions and runoff in Svalbard (1957-2018)"

_The Cryosphere, 2019_

## Referee Comment (RC1) · Anonymous Referee #1 · 3 May 2019

GENERAL

This is an exceptionally thorough and robust modelling-based paper investigating the climate mass balance (CMB), which includes surface and subsurface processes, across Svalbard between 1957 and 2018. It builds on previous similar work by the team, esp. first-authored papers by van Pelt, but this is the first time the latest version of the model (which now includes an improved subsurface scheme based on Marchenko et al., 2017b) has been applied to the whole of Svalbard. The model has a 1 km grid and is run at a 3 hourly time step and is therefore impressive in terms of its spatial and temporal resolution. The CMB is driven by downscaled climate data from

the High Resolution Limited Area Model (HIRLAM) regional climate model, which is forced by European Centre for Medium Range Weather Forecasts (ECMWF) reanalyses. This generates the meteorological forcing fields of air temperature, precipitation, cloud cover, relative humidity and air pressure.

The work uses an extensive data set of measurements to calibrate / validate the model (mass balance stake measurements from 8 glaciers; weather station data from 6 sites (4 on glacier; 2 off glacier); and shallow ice cores from 4 sites). These are listed in Table 1.

The calibration procedure is clearly explained and is logical and the principles have been discussed in two previous referenced papers. Here, parameters that are known to be sensitive are calibrated in sequence as described in section 3.2: 1. Two parameters affecting albedo are calibrated against net SW radiation data; 2. Two parameters in a function describing the downscaling of precipitation are calibrated against winter stake bass balance data; 3. Two parameters affecting summer melt are calibrated with observed summer balance data.

The fact that this model has a good history of being used in Svalbard and the fact that the RMSEs and biases after calibration are small, mean that the results will be the best that are currently available.

Results presented are quite extensive and informative and, as the authors state in the abstract, should be of value for scientists and practitioners interested in runoff to the oceans as well as ecologists interested in, for example, snow extent, duration and character (which has implications for reindeer grazing, for example).

The results / discussion section is focussed around a sequence of Figures showing: i) maps of mean conditions across Svalbard; ii) maps of trends over time (where significant); iii) time-series of spatially averaged trends in conditions. The consistency in the way the data are presented make the paper especially useful. The following results are shows and discussed: i) glacier CMB (Fig 5); ii) glacier ELA (Fig 6); iii) glacier firn pore

space in top 14 m (Fig 7a,b); iv) firn temperatures at 14 m (Fig 7 c,d); v) refreezing on and off glaciers (Fig. 9); vi) snow onset and disappearance dates off glaciers and across glacier ablation areas (Fig 10); vii) glacier and land runoff.

This represents a particularly impressive range of data sets presented and discussed from this type of modelling study.

The paper discusses sources of uncertainty throughout and has a synthesis section on this towards the end (section 4.6). Where results differ from those of similar previous work (but using earlier versions of the model, calibrated in different ways, run over different time periods, and across different spatial domains) the magnitudes and reasons for the discrepancies are revealed. The results and implications of the Svalbard work are also discussed in the context of similar work where appropriate in Arctic Canada and Greenland; this is especially the case when discussing the important finding of decreasing refreezing rates over time and therefore an increase in the likelihood of firn aquifers developing around the ELA.

So overall this represents excellent work by this team and shows the value of long-term monitoring but also the collection of shorter-term field measurements and their rigorous use in model development and application.

The work is exceptionally well presented in terms of the overall paper structure, as well as the clarity and precision of the writing, but also in the consistency and quality of the Figures.

SCIENTIFIC QUESTIONS / COMMENTS These are relatively minor:

As mentioned above, quoting from the paper, the meteorological forcing fields used to drive the CMB model are: air temperature, precipitation, cloud cover, relative humidity and air pressure. The answer is probably elsewhere in previous papers but a brief note on how these are used (together with other fields I assume) to calculate energy / mass balance at the surface would be useful. For example, there is no mention of

wind-speed here, and yet I assume this is required together with air temp and relative humidity to calculate the turbulent fluxes? And I assume theoretical clear sky solar radiation is used together with cloud cover to determine the incoming SW radiation?

P8 L17-19. The Bougamont et al (2015) work is for Greenland. How do you know that parameter values derived for the GrIS for t* are valid on Svalbard. The final sentence refers to the work on the GrIS I assume. Given the importance of albedo for melt and mass balance etc, some clarity is needed here about the validity of using the parameter values relevant for GrIS here in Svalbard. Is this a source of uncertainty that needs better recognition?

P9 L33&34. It's stated that the parameter Tsr has a strong impact on summer melt but most previous work has shown it's particularly important for winter accumulation. I can see it'll have an indirect impact on summer melt because of its direct impact on winter accumulation. Can you better justify why this parameter is tuned to the summer mass balance data and not the winter mass balance data?

P22 L1-2. There is a bit of confusion here as you seem to be discussing runoff rates due only to snow melt on land and comparing them to runoff rates due to snow and ice melt across glaciers. But, as you say later, runoff from land includes rainfall. Does runoff from glaciers also include rainfall? A better articulation of precisely how runoff is calculated for land and for glaciers is needed before the two values are compared. Can you separate out runoff from snow(ice) melt from runoff due to rainfall?

TYPOS / TECHNICAL ISSUES

Abstract P1 L4. Could say: "climatic mass balance (CMB) for the glaciers, snow conditions and runoff…"

L8. Suggest "small" not "weak"

P2 L4. "reveals" not "reveal"? The Longyearbyen time-series is singular not plural?

P2 L19. Could add the following reference to this list of previous studies here:

Rye, C.J., Willis, I.C., Arnold, N.S. and Kohler, J., 2012. On the need for automated multiobjective optimization and uncertainty estimation of glacier mass balance models. Journal of Geophysical Research: Earth Surface, v. 117,

P4 L21 "altitudes" (i.e. plural)

P5 Table 1. Table is not quite self-contained. Suggest adding to Table Heading and referring to Fig 1 heading for abbreviation names. Also to explain variables or say they're explained in the text.

P5 L10. Could add ref to Table 1 after final sentence here.

P6 L8 suggest "made" not "done"

P7 L5. Suggesting adding months when end of summer measurements are typically made (like April is stated earlier in the sentence for when Spring measurements are made). I'm guessing this is August or September (since 1 Sept. is stated as an average time below)?

P7 L5 Could delete "above described"

P11 L4 and Table 2. The term 'bias' is introduced here and referred to as "modelled minus observed". There are different definitions of bias so it might be worth clarifying precisely how it's defined here. Is it simply the Mean Absolute Difference (MAD)?

P11 L29. "five" should read 'six" here I assume? There are 6 sites mentioned in Table 1 and 3.

P11 L32. "...temperatures for both..."

P13 L11. Should this say "net CMB" to distinguish it from winter or summer that are also reported? Could clarify the first time you refer to net CMB, e.g. say "net CMB, hereafter just CMB..." or some such. In Abstract you might then also add the word "net"?

P17 L25-27. There is also some similar work to this reported recently from the Larsen C ice shelf, Antarctica that could also be compared / referenced. e.g.

Hubbard, B., Luckman, A., Ashmore, D.W., Bevan, S., Kulessa, B., Kuipers Munneke, P., Philippe, M., Jansen, D., Booth, A., Sevestre, H., Tison, J.L., O'Leary, M., and Rutt, I., 2016. Massive subsurface ice formed by refreezing of ice-shelf melt ponds. Nature Communications, 7.

Bevan, S. L., Luckman, A., Hubbard, B., Kulessa, B., Ashmore, D., Kuipers Munneke, P., O'Leary, M., Booth, A., Sevestre, H., and McGrath, D. 2017. Centuries of intense surface melt on Larsen C Ice Shelf, The Cryosphere, 11, 2743-2753.

P20 L3-5. There is a lack of clarity here. Here and the few sentences above need to better distinguish between a discussion of snow onset date and snow disappearance date. There's ambiguity here as it seems as though you might be comparing the trend in onset date (+1.4 days / decade) found in this study with trends in BOTH the onset date AND the disappearance date in a previous study. There is a bigger discrepancy in the disappearance date trends in the two studies than there is between the two onset date trends, and this probably needs stressing and discussing. I wouldn't say a disappearance date trend of +0.7 days / decade is comparable with 0 days per decade.

P23 L34. I think this should just read "...simulation, using the climate forcing...".

---

## Referee Comment (RC2) · Marco Möller (Referee) · 3 Jul 2019

Van Pelt et al. present a multi-decadal modeling study regarding snow and glacier mass balance on Svalbard that yielded results on a so-far unprecedented level of detail with respect to model resolution and captured processes. I very much congratulate the authors to this very thoroughly performed, documented and discussed modeling study which provides extremely valuable new knowledge to the field of Svalbard-wide glacier and snow research. I have no severe concerns regarding publication of this article. However, in its present form, the model description lacks a couple of important details that need to be added to the descriptions in order to make the methodology easier

to follow. In this respect, three substantial issues need special and more extensive attention, including limited additional data analysis. Taken together, I recommend to accept the manuscript of Van Pelt et al for publication in The Cryosphere after a minor revision along the issues outlined below.

Substantial comments:

1) P4L30f (& P9L16ff): I understand that you first linearly interpolate your 10km HIRLAM precipitation grid to the 1km resolution of your model. So far so good. However, in the next step you describe the application of a fixed linear fractional increase with elevation that you apply in addition. This step causes some concerns. I assume that the 10km elevation information in HIRLAM are not based on the S0 Terrengmodel Svalbard that you use in your mass balance model, right? This means that the average of the 1km elevations in your model across each 10km HIRLAM grid point and the elevation of this HIRLAM grid point itself do not equal each other. If this is the case, it introduces a physical inconsistency. Depending on the area altitude distribution of the 1km model grid points within each 10km HIRLAM grid point you either increase or decrease the total amount of precipitation that falls within this grid point by applying a fixed linear precipitation increase. Hence, the precipitation amounts which had been modeled by HIRLAM in a way that is physically consistent to synoptic forcing, are altered completely by your downscaling scheme. Moreover, this happens completely unstructured with respect to space, as the degree of alteration is only determined by the differences between the means of the 1km model elevations and the 10km HIRLAM elevations. I'm not sure if my interpretation above is what really happens; it could have also been a simple misunderstanding of your descriptions. In any case, I'd suggest that you comment on this issue in detail in the uncertainty discussion and/or revise your descriptions in the methods section accordingly to make them unambiguous in this respect.

2) P8L11ff: You implemented two novelties in your model. While the first one, the physically based percolation scheme, is fully referenced, the second one is not. How

were the parameters of your newly incorporated albedo scheme chosen? If you use a new or updated scheme, then you need to include information about how it was calibrated or how it is justified from a physical point of view. As you have various AWS data available I suppose that you could easily validate your new albedo scheme with shortwave radiation measurements from these stations. I do not ask for including an additional figure on this (even if it would be nice to have), but at least you should provide some comparative albedo numbers to validate the novelty in your model, especially its ability to produce a reliable course over the year. Your Svalbard-wide distribution of albedo values may also be compared to those modeled and described for 1979-2015 by Möller & Möller (2017) on the basis of MODIS data. This would yield important insights into the reliability of your so-far unreferenced albedo scheme.

3) P8L25ff: The application of RMSE minimization for finding the optimum values for K1 and K2 is certainly valid. However, I'd like to point towards a potential problem. In case the stake readings that are used as reference are not distributed equally with elevation, the RMSE minimization might not yield a proper combination of K1 and K2. This is because elevations with the highest number of stake readings are overrepresented and the minimization procedure thus concentrates on getting the accumulation at this specific elevation well while paying less attention to the other, underrepresented elevations. This issue has been detailed and documented for stake-based calibrations across Svalbard before by Möller et al. (2016) and you should at least account for it in the text. However, it would be better to check it in detail in order to avoid potentially wrong gradients or scaling coefficients that might lead to substantial under- and/or overestimation of climatic mass balance towards higher elevations. My concern is backed by the fact that the winter balances in Figure 3 clearly show, that modeled values tend to systematically underestimate the measured ones the more positive they become. If you visually place a linear fit to the blue points, the line would have a slope that for sure is distinctly larger than 1. I do not think that this issue will compromise your overall results as it probably only affects the most positive mass balances. Nevertheless, it needs to be presented in the uncertainty discussions section.

[Figure]

Detailed comments:

P2L19f: The reference to Day et al. (2012) is misleading here. They only calculate changes in surface mass balance on the basis of seasonal sensitivity characteristics. They neither use a mass balance model nor do they calculate absolute Svalbard-wide mass balance numbers. Instead of Day et al. and to complete the ensemble of Svalbard-wide mass balance calculations a reference to Möller et al. (2016) is missing here.

P2L34f: The references to Hagen et al. (2003) and Winther et al. (2003) are completely outdated as numerous studies related to seasonal snow coverage on Svalbard have been published since then, partly even with contributions of one or several co-authors of this study here. Hence, there is a need to include more up-to-date references here (e.g. Grabiec et al. 2011 or else).

P4L27: Include information about which ECMWF reanalysis products are used and over which periods. Even if this is partly deducible from the references it needs to be explicitly stated in the text.

P4L30f: Far more, especially quantitative, information about the applied lapse rates, increases and decays is needed here. The reader must fully understand what has been done without digging into previous literature.

P5L1: How did you calculate the significance of the trends? Information about this needs to be included here.

Figure 2: Just out of curiosity (as it is out of the scope of your study): do you have an explanation for the rather interesting pattern of precipitation trends visible in (d). I especially refer to the east coast of Wijdefjorden here.

P7L8: This value seems to be reasonable as it is quite often used for the transition from snow to firn. However, the choice appears to be rather arbitrary in the present form of the text. Information about how this choice was made, including appropriate

[Figure]

reference, needs to be given here. Moreover, as you give explicit information about the density applied to remaining snow you should also do so for other snow cover to ablation conversions that you consider in your model.

Figure 5b: One might think about the color scale here. People tend to associate blue colors with cooler conditions, which means more positive mass balances in glaciological studies. However, in the current version of this figure, blue represents a negative trend and thus a development towards more negative mass balances. Maybe it would be better skip the in the first instance ambiguous blue-red color scale here in favor of something more "uncommon". But that's just a suggestion as it reflects a rather subjective view.

Figure 9: An additional map showing the distribution of the percentage of melt and rainwater that is refrozen should be added here and any inferable information should to be included in the discussion where appropriate.

Figure 10: Same "problem" with the color scale as in Figure 5b. But this is still only a suggestion.

P22L21ff: You describe the usage of a fixed DEM and fixed glacier mask as a potential source of uncertainty and error. However, in the beginning of your paper you explicitly state that you calculate reference surface balances. Hence, your results do not suffer any "uncertainties" or "errors" due to the usage of fixed glacier extents and elevations. They simply represent a completely different quantity that is not comparable to "real" climatic mass balances which would be based on a time-varying glacier topography. This needs to be made clear in this section. You could of course keep the given descriptions, but treat them as deviations to what really happened on the glaciers and not as "uncertainties".

P23L7ff: The discussion of misestimations of precipitation fits to my substantial comment 3) in the beginning. The issue raised in this substantial comment needs to be included in this discussion, too.

P23L17ff: You might explicitly refer to the influences of wind-drifted snow, i.e. its potential to systematically increase or decrease local as well as regional accumulation rates. This information is certainly assumed in the sentence in question here, but it should be explicitly stated and referenced (e.g. Jaedicke and Gauer 2005; Grabiec et al. 2011).

References:

Grabiec M, Puczko D, Budzik T and Gajek G (2011), Snow distribution patterns on Svalbard glaciers derived from radio-echo soundings, Polish Polar Res., 32, 393-421.

Jaedicke C and Gauer P (2005), The influence of drifting snow on the location of glaciers on western Spitsbergen, Svalbard, Ann. Glaciol., 42, 237-242.

Möller M and Möller R (2017), Modeling glacier‐surface albedo across Svalbard for the 1979–2015 period: The HiRSvaC500‐$\alpha$ data set, J. Adv. Model. Earth Syst., 9, 404–422, doi:10.1002/2016MS000752.

Möller M, Obleitner F, Reijmer CH, Pohjola VA, Głowacki P and Kohler J (2016), Adjustment of regional climate model output for modeling the climatic mass balance of all glaciers on Svalbard, J. Geophys. Res. Atmos., 121, 5411-5429, doi: 10.1002/2015JD024380.

---

## Author Comment (AC1) · 11 Jul 2019

**Response to Anonymous Referee #1**

*RC = Referee Comment*
*AR = Author Response*

RC: This is an exceptionally thorough and robust modelling-based paper investigating the climate mass balance (CMB), which includes surface and subsurface processes, across Svalbard between 1957 and 2018. It builds on previous similar work by the team, esp. first-authored papers by van Pelt, but this is the first time the latest version of the model (which now includes an improved subsurface scheme based on Marchenko et al., 2017b) has been applied to the whole of Svalbard. The model has a 1 km grid and is run at a 3 hourly time step and is therefore impressive in terms of its spatial and temporal resolution. The CMB is driven by downscaled climate data from the High Resolution Limited Area Model (HIRLAM) regional climate model, which is forced by European Centre for Medium Range Weather Forecasts (ECMWF) reanalyses. This generates the meteorological forcing fields of air temperature, precipitation, cloud cover, relative humidity and air pressure. The work uses an extensive data set of measurements to calibrate / validate the model (mass balance stake measurements from 8 glaciers; weather station data from 6 sites(4 on glacier; 2 off glacier); and shallow ice cores from 4 sites). These are listed in Table 1.The calibration procedure is clearly explained and is logical and the principles have been discussed in two previous referenced papers. Here, parameters that are known to be sensitive are calibrated in sequence as described in section 3.2: 1. Two parameters affecting albedo are calibrated against net SW radiation data; 2. Two parameters in a function describing the downscaling of precipitation are calibrated against winter stake bass balance data; 3. Two parameters affecting summer melt are calibrated with observed summer balance data. The fact that this model has a good history of being used in Svalbard and the fact that the RMSEs and biases after calibration are small, mean that the results will be the best that are currently available. Results presented are quite extensive and informative and, as the authors state in the abstract, should be of value for scientists and practitioners interested in runoff to the oceans as well as ecologists interested in, for example, snow extent, duration and character (which has implications for reindeer grazing, for example).The results / discussion section is focused around a sequence of Figures showing: i)maps of mean conditions across Svalbard; ii) maps of trends over time (where significant); iii) time-series of spatially averaged trends in conditions. The consistency in the way the data are presented make the paper especially useful. The following results are shown and discussed: i) glacier CMB (Fig 5); ii) glacier ELA (Fig 6); iii) glacier firn pore space in top 14 m (Fig 7a,b); iv) firn temperatures at 14 m (Fig 7 c,d); v) refreezing on and off glaciers (Fig. 9); vi) snow onset and disappearance dates off glaciers and across glacier ablation areas (Fig 10); vii) glacier and land runoff. This represents a particularly impressive range of data sets presented and discussed from this type of modelling study. The paper discusses sources of uncertainty throughout and has a synthesis section on this towards the end (section 4.6). Where results differ from those of similar previous work (but using earlier versions of the model, calibrated in different ways, run over different time periods, and across different spatial domains) the magnitudes and reasons for the discrepancies are revealed. The results and implications of the Svalbard work are also discussed in the context of similar work where appropriate in Arctic Canada and Greenland; this is especially the case when discussing the important finding of decreasing refreezing rates over time and therefore an increase in the likelihood of firn aquifers developing around the ELA. So overall this represents excellent work by this team and shows the value of long-term monitoring but also the collection of shorter-term field measurements and their rigorous use in model development and application. The work is exceptionally well presented in terms of the overall paper structure, as well as the clarity and precision of the writing, but also in the consistency and quality of the Figures.

AR: We are very grateful for this very positive feedback! And we thank the reviewer for the useful comments, which we address below, and which have helped to improve the manuscript.

RC: As mentioned above, quoting from the paper, the meteorological forcing fields used to drive the CMB model are: air temperature, precipitation, cloud cover, relative humidity and air pressure. The answer is probably elsewhere in previous papers but a brief note on how these are used (together with other fields I assume) to calculate energy/ mass balance at the surface would be useful. For example, there is no mention of wind-speed here, and yet I assume this is required together with air temp and relative humidity to calculate the turbulent fluxes? And I assume theoretical clear sky solar radiation is used together with cloud cover to determine the incoming SW radiation?

AR: More details on the individual energy balance components and their dependence on climate fields is given in Van Pelt et al. (2012). However, we agree some more information would be helpful to include here as well, so we added the following in Sect. 3.1:

*"Solving the surface energy balance requires input of near-surface meteorological conditions, including air temperature, cloudiness, relative humidity, air pressure and precipitation (Van Pelt et al. 2012). No wind information is needed since sensible and latent heat exchange depend solely on near-surface temperature and specific humidity gradients, following katabatic turbulent exchange relations by Oerlemans & Grisogono (2002)."*

RC: P8 L17-19. The Bougamont et al (2015) work is for Greenland. How do you know that parameter values derived for the GrIS for t* are valid on Svalbard. The final sentence refers to the work on the GrIS I assume. Given the importance of albedo for melt and mass balance etc, some clarity is needed here about the validity of using the parameter values relevant for GrIS here in Svalbard. Is this a source of uncertainty that needs better recognition?

AR: This a good point. At present we cannot confirm or deny that the t* values from Bougamont et al. (2005) are appropriate for Svalbard glaciers. Although we did not calibrate t* values, it is worth mentioning that as part of the calibration procedure in Section 3.2 we have used SW net observations from three AWSs in Svalbard to calibrate fresh snow albedo and the minimum snowfall amount at which the snow albedo is reset to the fresh snow albedo. These are two albedo parameters to which modelled melt has in a previous study (Van Pelt et al. 2012) been shown to be highly sensitive. By using SWnet data in the calibration, we avoid substantial biases in the surface energy balance (and calculated melt rates) resulting from potentially inaccurate parameter values affecting incoming and reflected SW radiation, including t*. However, any inaccuracies in chosen values of, for example, t* would be compensated for by a potentially different value for the fresh snow albedo and/or the snowfall threshold. In future work, a more detailed comparison of modelled and observed albedo at multiple sites in Svalbard and for longer time-series would allow for more extensive calibration of albedo parameters. To acknowledge uncertainty in parameters like t*, we now include the following sentences in Sect. 4.6:

*"Energy balance parameters were taken as in the aforementioned studies, with the exception of the fresh snow albedo ($\alpha_{fs}$), the associated minimum snowfall threshold ($P_{th}$), and the background turbulent exchange coefficient ($C_b$), which were calibrated against observational data (Sect. 3.2). The new albedo scheme assumes that previously used values of t* for Greenland (Bougamont et al. 2005) are also applicable to Svalbard. Potential inaccuracies in parameters like t* will introduce uncertainty in modeled albedo values, as it introduces compensating errors in calibrated parameters; in the case of t*, compensating errors would arise in $\alpha_{fs}$ and $P_{th}$. However, the calibration procedure assures that, despite compensating errors, net biases in relevant model output, e.g. melt, is minimized. More careful calibration of albedo parameters, including t*, is planned for future work using a more extensive dataset of albedo measurements across Svalbard."*

Also, in response to reviewer #2, we have additionally extended the description of albedo in Sect. 3.1, to better introduce the albedo scheme and give more information on where the parameter values come from (if not calibrated).

RC: P9 L33&34. It's stated that the parameter Tsr has a strong impact on summer melt but most previous work has shown it's particularly important for winter accumulation. I can see it'll have an indirect impact on summer melt because of its direct impact on winter accumulation. Can you better justify why this parameter is tuned to the summer mass balance data and not the winter mass balance data?

AR: Indeed, Tsr has some impact on winter balance as well. However, we find that the sensitivity of the winter balance to Tsr changes is about twenty (!) times smaller than the sensitivity of summer balance to Tsr changes for the mass balance stake locations. Several factors play a role here, but the relatively insensitivity of winter balance to Tsr is primarily explained by the fact that rainfall during the core winter season is (still) rare in Svalbard, especially at higher elevations. And in case any rain falls during the core winter season, most of the rain water will refreeze in the snow pack thereby not inducing any runoff. The significant impact of Tsr on melt and runoff (and thereby summer balance) has previously been quantified in Van Pelt et al. (2012) for Nordenskiöldbreen. We have added a sentence to Sect. 3.2 explaining the relative insensitivity of winter balance to Tsr.

RC: P22 L1-2. There is a bit of confusion here as you seem to be discussing runoff rates due only to snow melt on land and comparing them to runoff rates due to snow and ice melt across glaciers. But, as you say later, runoff from land includes rainfall. Does runoff from glaciers also include rainfall? A better articulation of precisely how runoff is calculated for land and for glaciers is needed before the two values are compared. Can you separate out runoff from snow(ice) melt from runoff due to rainfall?

AR: We now try to avoid this confusion by first giving the definition of runoff in Sect. 4.5:

*"Here, runoff refers to the amount of water originating from melt and rainfall at the surface and available at the base of the snow/firn pack (if present) or ice/soil surface after accounting for retention by refreezing and irreducible water storage."*

**Typos / technical issues**

RC: Abstract P1 L4. Could say: "climatic mass balance (CMB) for the glaciers, snow conditions and runoff"
RC: L8. Suggest "small" not "weak"
RC: P2 L4. "reveals" not "reveal"? The Longyearbyen time-series is singular not plural?
RC: P2 L19. Could add the following reference to this list of previous studies here:
Rye, C.J., Willis, I.C., Arnold, N.S. and Kohler, J., 2012. On the need for automated multiobjective optimization and uncertainty estimation of glacier mass balance models. Journal of Geophysical Research: Earth Surface, v. 117,
RC: P4 L21 "altitudes" (i.e. plural)

AR: All fixed.

RC: P5 Table 1. Table is not quite self-contained. Suggest adding to Table Heading and referring to Fig 1 heading for abbreviation names. Also to explain variables or say they're explained in the text.

AR: We have extended the Table header with more details about the used abbreviations.

RC: P5 L10. Could add ref to Table 1 after final sentence here.
RC: P6 L8 suggest "made" not "done"
RC: P7 L5. Suggesting adding months when end of summer measurements are typically made (like April is stated earlier in the sentence for when Spring measurements are made). I'm guessing this is August or September (since 1 Sept. is stated as an average
time below)?
RC: P7 L15 Could delete "above described"

AR: All fixed.

RC: P11 L4 and Table 2. The term 'bias' is introduced here and referred to as "modelled minus observed". There are different definitions of bias so it might be worth clarifying precisely how it's defined here. Is it simply the Mean Absolute Difference (MAD)?

AR: We have added that the bias is the mean absolute difference.

RC: P11 L29. "five" should read 'six" here I assume? There are 6 sites mentioned in Table 1 and 3.
RC: P11 L32. "…temperatures for both…"

AR: Both fixed.

RC: P13 L11. Should this say "net CMB" to distinguish it from winter or summer that are also reported? Could clarify the first time you refer to net CMB, e.g. say "net CMB, hereafter just CMB…" or some such. In Abstract you might then also add the word "net"?

AR: Good point. We have followed these suggestions.

RC: P17 L25-27. There is also some similar work to this reported recently from the Larsen C ice shelf, Antarctica that could also be compared / referenced. e.g.

Hubbard, B., Luckman, A., Ashmore, D.W., Bevan, S., Kulessa, B., Kuipers Munneke, P., Philippe, M., Jansen, D., Booth, A., Sevestre, H., Tison, J.L., O'Leary, M., and Rutt, I., 2016. Massive subsurface ice formed by refreezing of ice-shelf melt ponds. Nature Communications, 7.

Bevan, S. L., Luckman, A., Hubbard, B., Kulessa, B., Ashmore, D., Kuipers Munneke, P., O'Leary, M., Booth, A., Sevestre, H., and McGrath, D. 2017. Centuries of intense surface melt on Larsen C Ice Shelf, The Cryosphere, 11, 2743-2753.

AR: References added.

RC: P20 L3-5. There is a lack of clarity here. Here and the few sentences above need to better distinguish between a discussion of snow onset date and snow disappearance date. There's ambiguity here as it seems as though you might be comparing the trend in onset date (+1.4 days / decade) found in this study with trends in BOTH the onset date AND the disappearance date in a previous study. There is a bigger discrepancy in the disappearance date trends in the two studies than there is between the two onset date trends, and this probably needs stressing and discussing. I wouldn't say a disappearance date trend of +0.7 days / decade is comparable with 0 days per decade.

AR: We have reformulated the associated sentences to improve clarity.

RC: P23 L34. I think this should just read "…simulation, using the climate forcing…".

AR: Fixed.

---

## Author Comment (AC2) · 11 Jul 2019

**Response to reviewer #2 (Marco Möller)**

*RC = Referee Comment*
*AR = Author Response*

RC: Van Pelt et al. present a multi-decadal modeling study regarding snow and glacier mass balance on Svalbard that yielded results on a so-far unprecedented level of detail with respect to model resolution and captured processes. I very much congratulate the authors to this very thoroughly performed, documented and discussed modeling study which provides extremely valuable new knowledge to the field of Svalbard-wide glacier and snow research. I have no severe concerns regarding publication of this article. However, in its present form, the model description lacks a couple of important details that need to be added to the descriptions in order to make the methodology easier to follow. In this respect, three substantial issues need special and more extensive attention, including limited additional data analysis. Taken together, I recommend to accept the manuscript of Van Pelt et al for publication in The Cryosphere after a minor revision along the issues outlined below.

AR: We thank the reviewer for the very positive feedback! We are also grateful for the substantial and detailed comments, which have helped us to improve the manuscript.

**Substantial comments:**

RC: 1) P4L30f (& P9L16ff): I understand that you first linearly interpolate your 10km HIRLAM precipitation grid to the 1km resolution of your model. So far so good. However, in the next step you describe the application of a fixed linear fractional increase with elevation that you apply in addition. This step causes some concerns. I assume that the 10km elevation information in HIRLAM are not based on the S0 Terrengmodel Svalbard that you use in your mass balance model, right? This means that the average of the 1km elevations in your model across each 10km HIRLAM grid point and the elevation of this HIRLAM grid point itself do not equal each other. If this is the case, it introduces a physical inconsistency. Depending on the area altitude distribution of the 1km model grid points within each 10km HIRLAM grid point you either increase or decrease the total amount of precipitation that falls within this grid point by applying a fixed linear precipitation increase. Hence, the precipitation amounts which had been modeled by HIRLAM in a way that is physically consistent to synoptic forcing, are altered completely by your downscaling scheme. Moreover, this happens completely unstructured with respect to space, as the degree of alteration is only determined by the differences between the means of the 1km model elevations and the 10km HIRLAM elevations. I'm not sure if my interpretation above is what really happens; it could have also been a simple misunderstanding of your descriptions. In any case, I'd suggest that you comment on this issue in detail in the uncertainty discussion and/or revise your descriptions in the methods section accordingly to make them unambiguous in this respect.

AR: We believe we understand the reviewer's concern, but do not regard the precipitation - elevation correction method as physically inconsistent. The two reasons to apply the precipitation – elevation equation are to 1) account for any elevation differences between the detailed 1-km DEM and the coarser 10-km HIRLAM DEM (interpolated to 1-km resolution), and 2) to correct for any Svalbard-wide biases between modelled and observed precipitation. The former is controlled by K2, while the latter is controlled by K1. This precipitation adjustment is not 'mass-conserving', since we correct for biases between model and observations (K1 is not equal to 1). Furthermore, as suggested by the reviewer, there may be (small) elevation deviations between the mean of 1-km height values and the corresponding regional climate model elevation value. This is not at all problematic, since we in fact desire that our correction method also compensates for these height errors by altering the precipitation amount. The actual 1-km DEM will contain more detail than the interpolated HIRLAM-DEM; any positive deviation of the surface height will lead to a positive correction of the local precipitation, while a negative height deviation will lead to a negative precipitation correction. The main advantage of this downscaling is that we introduce the effect of orographic lifting on precipitation at scales

smaller than the 10-km spatial resolution of the regional climate model. To clarify our approach, we have added the following in Sect. 3.2:

*"Values from the 1-km DEM (z) will contain more detail than the z0 values interpolated from the coarser regional climate model grid; any positive deviation of the surface height (z-z0>0) will lead to a positive correction of the local precipitation, while a negative height deviation (z-z0<0) will lead to a negative precipitation correction. With this approach, we account for the effect of local topography on precipitation, thereby capturing the impact of orographic lifting at scales smaller than the resolution of the regional climate model. In addition to compensation for biases in modelled precipitation (by calibrating $K_1$) potential surface height discrepancies at spatial scales of 10-km and greater that may arise from the use of a different DEM in the regional climate model are also automatically compensated for.*

RC: 2) P8L11ff: You implemented two novelties in your model. While the first one, the physically based percolation scheme, is fully referenced, the second one is not. How were the parameters of your newly incorporated albedo scheme chosen? If you use a new or updated scheme, then you need to include information about how it was calibrated or how it is justified from a physical point of view. As you have various AWS data available, I suppose that you could easily validate your new albedo scheme with shortwave radiation measurements from these stations. I do not ask for including an additional figure on this (even if it would be nice to have), but at least you should provide some comparative albedo numbers to validate the novelty in your model, especially its ability to produce a reliable course over the year. Your Svalbard-wide distribution of albedo values may also be compared to those modeled and described for 1979-2015 by Möller & Möller (2017) on the basis of MODIS data. This would yield important insights into the reliability of your so-far unreferenced albedo scheme.

AR: This is a valid point. A similar comment was given by reviewer #1, who questioned the transferability of albedo parameters (t*) from Greenland to Svalbard. We agree that some more information about the albedo scheme is needed. In Sect. 3.1, we have now extended the description of the albedo scheme by explaining what parameter values were chosen and based on which references. First of all, we refer to Bougamont et al. (2005) for the characteristic decay time-scale (t*) values, which were optimized on Greenland and used here as well. In the uncertainty discussion (Sect. 4.6), we have now added a discussion on uncertainty related to the potential errors in chosen albedo parameter values like t* (in response to reviewer #1). Secondly, for three other albedo parameters (albedo of ice [0.39], albedo of firn [0.52] and characteristic snow depth [7 mm w.e.]) we used values that have been calibrated in a previous study with the model (Van Pelt and Kohler, 2015). This information is now added to Sect. 3.1. Thirdly, regarding calibration/validation of the new albedo scheme, it is worth mentioning that we optimize two key albedo parameters (fresh snow albedo & snowfall threshold to reset albedo to the fresh snow albedo) against observations of net SW radiation (see Sect. 3.2). By doing so any biases in net SW radiation resulting from the new albedo scheme are minimized. A sentence has been added to Sect. 3.1 to clarify this. We do not think it is needed to include a more detailed validation or comparison of albedo time-series in this study, although it would be useful to do this in future work with a more specific focus on albedo (this is now also discussed in Sect. 4.6).

The new description of albedo in Sect. 3.1 reads:

*"Additionally, we have extended the original snow age and snow depth dependent albedo scheme (Oerlemans et al. 1998). The original fixed characteristic time-scale for exponential decay of snow albedo due to ageing has been replaced with a temperature dependent time-scale (t*). As in Bougamont et al. (2005) snow albedo decays fastest when the surface is melting (t* = 15 d), and for dry snow t* linearly increases from 30 to 100 days between 0 and -10 oC. The updated albedo scheme avoids overestimation of the albedo of melting surfaces in the early melt season. Other albedo parameters, including the albedo of ice (0.39), albedo of firn (0.52), and the characteristic snow depth for albedo decay of thin snow covers (7 mm w.e.) were taken as in (Van Pelt and Kohler, 2015). To avoid potential systematic biases resulting from the new albedo scheme, we have included the*

*fresh snow albedo ($\alpha_{fs}$) minimum snowfall threshold used to reset the snow albedo to the fresh snow albedo ($P_{th}$) in the calibration process, as described in Sect. 3.2."*

The new discussion about uncertainty in albedo parameters in Sect. 4.6 reads:

*"Energy balance parameters were taken as in the aforementioned studies, with the exception of the fresh snow albedo ($\alpha_{fs}$), the associated minimum snowfall threshold ($P_{th}$), and the background turbulent exchange coefficient ($C_b$), which were calibrated against observational data (Sect. 3.2). The new albedo scheme assumes that previously used values of t\* for Greenland (Bougamont et al. 2005) are also applicable to Svalbard. Potential inaccuracies in parameters like t\* will introduce uncertainty in modeled albedo values, as it introduces compensating errors in calibrated parameters; in the case of t\*, compensating errors would arise in $\alpha_{fs}$ and $P_{th}$. However, the calibration procedure assures that, despite compensating errors, net biases in relevant model output, e.g. melt and runoff, are minimized. More careful calibration of albedo parameters, including t\*, is planned for future work using a more extensive dataset of albedo measurements across Svalbard."*

Regarding the albedo maps that were presented in Möller and Möller (2017) we are happy to share maps with the reviewer (and the data behind them) that can be used for comparison. When writing this manuscript, we have carefully selected the variables that we thought were of most value to present and albedo was in the end not selected. We decided to have more focus on the mass balance components and stay away from a detailed analysis of individual energy balance components, including albedo.

RC: 3) P8L25ff: The application of RMSE minimization for finding the optimum values for K1 and K2 is certainly valid. However, I'd like to point towards a potential problem.
In case the stake readings that are used as reference are not distributed equally with elevation, the RMSE minimization might not yield a proper combination of K1 and K2.
This is because elevations with the highest number of stake readings are overrepresented and the minimization procedure thus concentrates on getting the accumulation at this specific elevation well while paying less attention to the other, underrepresented elevations. This issue has been detailed and documented for stake-based calibrations across Svalbard before by Möller et al. (2016) and you should at least account for it in the text. However, it would be better to check it in detail in order to avoid potentially wrong gradients or scaling coefficients that might lead to substantial under and/ or overestimation of climatic mass balance towards higher elevations. My concern is backed by the fact that the winter balances in Figure 3 clearly show, that modeled values tend to systematically underestimate the measured ones the more positive they become. If you visually place a linear fit to the blue points, the line would have a slope that for sure is distinctly larger than 1. I do not think that this issue will compromise your overall results as it probably only affects the most positive mass balances. Nevertheless, it needs to be presented in the uncertainty discussions section.

AR: This is an interesting comment. We will start our response by showing a figure. To assess whether there may be a potential bias increasing with elevation we have made height profiles of $b_w$, $b_s$ and $b_n$ for the eight observed glaciers. Here data are averaged in 100 m elevation bins and over the whole observation period.

[Figure]

What can be seen is that there are some glaciers (particularly KNG and HBR) where $b_w$ biases indeed increase with elevation. However, for the glaciers with the highest elevations, i.e. over 1000 m a.s.l. (HDF and NBR), we do not see this effect. What this shows is that the bias for high observed $b_w$ values that seems apparent in Fig. 3 in the manuscript is likely to come from underestimated precipitation at the highest stakes on KNG and HBR, where long data records exist (i.e. many data points) and precipitation amounts are among the highest on Svalbard (much higher than at the much higher-located stakes on NBR and HDF). These local effects are apparently not properly captured by the model, but it cannot be concluded that precipitation amounts are off at high elevations in general. We now refer to Möller et al. (2016) and have extended the discussion on this at the end of Sect 3.2:

*"On the other hand, underestimation of $b_w$ is apparent for KNG and HBR (Fig. 3, Table 2), which results from underestimated orographic precipitation at high elevations on these glaciers. Nevertheless, high-elevation biases of $b_w$ do not arise on the only two glaciers extending above 1000 m a.s.l., which indicates that the $b_w$ offsets on KNG and HBR are not a systematic feature for high elevation sites in general. The relative lack of stake observations at heights above 1000 m a.s.l. implies increased uncertainty of modelled precipitation estimates at these elevations (Möller et al. 2016)".*

**Detailed comments:**

RC: P2L19f: The reference to Day et al. (2012) is misleading here. They only calculate changes in surface mass balance on the basis of seasonal sensitivity characteristics. They neither use a mass balance model nor do they calculate absolute Svalbard wide mass balance numbers. Instead of Day et al. and to complete the ensemble of Svalbard-wide mass balance calculations a reference to Möller et al. (2016) is missing here.

AR: We have added a reference to Möller et al. (2016) and removed the reference to Day et al. (2012).

RC: P2L34f: The references to Hagen et al. (2003) and Winther et al. (2003) are completely outdated as numerous studies related to seasonal snow coverage on Svalbard have been published since then, partly even with contributions of one or several co-authors of this study here. Hence, there is a need to include more up-to-date references here (e.g. Grabiec et al. 2011 or else).

AR: We believe the two 2003 references are still relevant, but now also include the suggested newer reference to Grabiec et al. (2011).

RC: P4L27: Include information about which ECMWF reanalysis products are used and over which periods. Even if this is partly deducible from the references it needs to be explicitly stated in the text.

AR: Fixed.

RC: P4L30f: Far more, especially quantitative, information about the applied lapse rates, increases and decays is needed here. The reader must fully understand what has been done without digging into previous literature.

AR: This is a good point. We have added the following sentences in Sect. 2.1:

*"Elevation functions for temperature and air pressure were constructed per 3h time-step through respectively linear and exponential regression of the regional climate model values and their corresponding elevations; this procedure was repeated for blocks of 4x4 grid cells and regression coefficients were averaged for the whole grid to obtain a single lapse-rate for temperature and exponential decay coefficient for air pressure per time-step."*

RC: P5L1: How did you calculate the significance of the trends? Information about this needs to be included here.

AR: It is now added that significance means that a zero trend is not included in the 2-sigma confidence bounds of a trend.

RC: Figure 2: Just out of curiosity (as it is out of the scope of your study): do you have an explanation for the rather interesting pattern of precipitation trends visible in (d). I especially refer to the east coast of Wijdefjorden here.

AR: Well, the precipitation trends are generally very small; the trend values that still appear as colors (non-gray) in the map are also only just significant. In general a slight precipitation increase in Svalbard has also been observed in observational records (see also Introduction). And the somewhat stronger precipitation trends in northern Svalbard could be a result of retreating sea ice having a larger impact on moisture availability there. This effect has also been suggested to amplify in a future climate (see for example the recent Climate in Svalbard 2100 report).

RC: P7L8: This value seems to be reasonable as it is quite often used for the transition from snow to firn. However, the choice appears to be rather arbitrary in the present form of the text. Information about how this choice was made, including appropriate reference, needs to be given here. Moreover, as you give explicit information about the density applied to remaining snow you should also do so for other snow cover to ablation conversions that you consider in your model.

AR: We have removed the 550 kg m$^{-3}$ value from the paper and instead refer to Van Pelt et al. (2018) where the summer balance calculation from stake measurements (in that case on Nordenskiöldbreen) is also described. For the conversion of stake heights to winter balance estimates, fresh snow density is needed, which is based on snow pit data. Since this is already described in Van Pelt et al. (2016) we only keep the reference to that paper for more information.

RC: Figure 5b: One might think about the color scale here. People tend to associate blue colors with cooler conditions, which means more positive mass balances in glaciological studies. However, in the

current version of this figure, blue represents a negative trend and thus a development towards more negative mass balances. Maybe it would be better skip the in the first instance ambiguous blue-red color scale here in favor of something more "uncommon". But that's just a suggestion as it reflects a rather subjective view.
RC: Figure 10: Same "problem" with the color scale as in Figure 5b. But this is still only a suggestion.

AR: Thanks for the suggestion. We understand the confusion this may cause, but we still think it is good to stick to the current choice of colormap for the sake of consistency. We like to be consistent throughout the manuscript with blue colors for a negative trend and red colors for a positive trend. For some variables this may be intuitive (e.g. temperature), whereas for others like CMB it may be counterintuitive, but this seems unavoidable.

RC: Figure 9: An additional map showing the distribution of the percentage of melt and rainwater that is refrozen should be added here and any inferable information should to be included in the discussion where appropriate.

AR: This is a useful suggestion, and we have decided to add two panels to Fig. 9 showing the fraction of melt and rainfall that refreezes (Fig. 9c) and the associated trends (Fig. 9d). These new results interestingly show that highest refrozen fraction values occur on Lomonosovfonna and lowest values in coastal regions in southern Svalbard. Furthermore, it is found that no sites experience a refrozen fraction that is close to 1 (values up to 0.8 are found), which indirectly shows the absence of cold firn in Svalbard during the simulation period. Trends of the refrozen fraction reveal most negative values in northern Svalbard. All this is now discussed in more detail in the second and third paragraph in Sect. 4.3.

RC: P22L21ff: You describe the usage of a fixed DEM and fixed glacier mask as a potential source of uncertainty and error. However, in the beginning of your paper you explicitly state that you calculate reference surface balances. Hence, your results do not suffer any "uncertainties" or "errors" due to the usage of fixed glacier extents and elevations.
They simply represent a completely different quantity that is not comparable to "real" climatic mass balances which would be based on a time-varying glacier topography. This needs to be made clear in this section. You could of course keep the given descriptions, but treat them as deviations to what really happened on the glaciers and not as "uncertainties".

AR: Good point. We now refer to 'deviations' rather than 'uncertainties' instead, and have added some text on the use of "reference" surfaces for mass balance modelling (Sect. 4.6).

RC: P23L7ff: The discussion of misestimations of precipitation fits to my substantial comment 3) in the beginning. The issue raised in this substantial comment needs to be included in this discussion, too.

AR: We have added some discussion on this now earlier on in Sect. 3.2. For more details, see our reply to substantial comment 3).

RC: P23L17ff: You might explicitly refer to the influences of wind-drifted snow, i.e. its potential to systematically increase or decrease local as well as regional accumulation rates.
This information is certainly assumed in the sentence in question here, but it should be explicitly stated and referenced (e.g. Jaedicke and Gauer 2005; Grabiec et al. 2011).

AR: We have added a notion on wind-driven snow redistribution and included the additional references (Sect. 4.6).